# Fucosylation and protein glycosylation create functional receptors for cholera toxin

Amberlyn M Wands[1†], Akiko Fujita[1†], Janet E McCombs[1], Jakob Cervin[2,3], Benjamin Dedic[4], Andrea C Rodriguez[1], Nicole Nischan[1], Michelle R Bond[1‡], Marcel Mettlen[5], David C Trudgian[1], Andrew Lemoff[1], Marianne Quiding-Järbrink[2,3], Bengt Gustavsson[6], Catharina Steentoft[7,8], Henrik Clausen[7,8], Hamid Mirzaei[1], Susann Teneberg[2,4], Ulf Yrlid[2,3*], Jennifer J Kohler[1*]

[1]Department of Biochemistry, University of Texas Southwestern Medical Center, Dallas, United States; [2]Department of Microbiology and Immunology, Institute of Biomedicine, University of Gothenburg, Gothenburg, Sweden; [3]Mucosal Immunobiology and Vaccine Center, Institute of Biomedicine, University of Gothenburg, Gothenburg, Sweden; [4]Department of Biochemistry and Cell Biology, Institute of Biomedicine, Sahlgrenska Academy, University of Gothenburg, Gothenburg, Sweden; [5]Department of Microbiology and Immunology, Institute of Biomedicine, University of Gothenburg, Gothenburg, Sweden; [6]Department of Surgery, University of Gothenburg, Gothenburg, Sweden; [7]Copenhagen Center for Glycomics, Department of Cellular and Molecular Medicine, Faculty of Health Sciences, University of Copenhagen, Copenhagen, Denmark; [8]School of Dentistry, Faculty of Health Sciences, University of Copenhagen, Copenhagen, Denmark

*For correspondence: ulf.yrlid@microbio.gu.se (UY); jennifer.kohler@utsouthwestern.edu (JJK)

[†]These authors contributed equally to this work

Present address: [‡]National Institute of Diabetes and Digestive and Kidney Diseases, National Institutes of Health, Bethesda, United States

Competing interests: The authors declare that no competing interests exist.

**Abstract** Cholera toxin (CT) enters and intoxicates host cells after binding cell surface receptors using its B subunit (CTB). The ganglioside (glycolipid) GM1 is thought to be the sole CT receptor; however, the mechanism by which CTB binding to GM1 mediates internalization of CT remains enigmatic. Here we report that CTB binds cell surface glycoproteins. Relative contributions of gangliosides and glycoproteins to CTB binding depend on cell type, and CTB binds primarily to glycoproteins in colonic epithelial cell lines. Using a metabolically incorporated photocrosslinking sugar, we identified one CTB-binding glycoprotein and demonstrated that the glycan portion of the molecule, not the protein, provides the CTB interaction motif. We further show that fucosylated structures promote CTB entry into a colonic epithelial cell line and subsequent host cell intoxication. CTB-binding fucosylated glycoproteins are present in normal human intestinal epithelia and could play a role in cholera.

## Introduction

The bacterium *Vibrio cholerae* is the etiological agent of cholera (*Foster and Baron, 1996*). Cholera toxin (CT) is secreted by *V. cholerae* and is the direct cause of the profuse, watery diarrhea that characterizes fatal cholera. CT is a heterohexamer comprising one copy of cholera toxin subunit A (CTA) and five copies of subunit B (CTB). Mechanistic studies have yielded the following model for how CT intoxicates host cells (*Sánchez and Holmgren, 2008*; *Lencer, 2003*). The CTB subunits of the holotoxin bind receptors on the surface of host enterocytes, enabling endocytosis of CT. CT follows a

**eLife digest** Cholera is a serious diarrheal disease that can be deadly if left untreated. It is caused by eating food, or drinking water, contaminated by the bacterium *Vibrio cholerae*. This bacterium can survive passage through the acidic conditions of the stomach. Inside the small intestine, *V. cholerae* attaches to the intestinal wall and starts producing cholera toxin. The toxin enters intestinal cells, causing them to release water and ions, including sodium and chloride ions. The salt-water environment created inside the intestine can, by osmosis, draw up to a further six liters of water into the intestine each day. This results in the copious production of watery diarrhea and severe dehydration.

Cholera toxin is composed of six protein subunits, including five copies of cholera toxin subunit B (CTB). CTB subunits help the uptake of the toxin by intestinal cells, and it has long been reported that CTB subunits attach to intestinal cells by binding to a cell surface molecule called GM1. CTB subunits have a high affinity for GM1, yet recent work suggests CTB may not bind exclusively to GM1; one or more additional cell surface molecules may be directly involved in cholera toxin uptake.

Wands et al. now reveal that numerous cell surface molecules are recognized by CTB, and that these molecules can assist cholera toxin uptake by host cells. Glycoproteins, proteins that are marked with sugar molecules, were shown to be the primary CTB binding sites on human colon cells, and it was the glycoprotein's sugar component, not the protein itself, that interacted with CTB. Wands et al. discovered that in particular glycoproteins containing a sugar called fucose were largely responsible for CTB binding and toxin uptake. Together these findings reveal a previously unrecognized mechanism for cholera toxin entry into host cells, and suggest that fucose-containing or fucose-mimicking molecules could be developed as new treatments for cholera.

retrograde trafficking pathway to the ER where it is disassembled to release CTA. CTA enters the cytoplasm and catalyzes ADP-ribosylation of the α-subunits of heterotrimeric GTP-binding proteins ($G\alpha_s$). The resulting extended activation of $G\alpha_s$ leads to increased activity of adenylate cyclase, raising intracellular cAMP levels. Elevated cAMP causes activation of chloride channels and chloride efflux, followed by massive secretion of water and ions into the intestinal lumen. Affected individuals can experience rapid and severe dehydration, sometimes leading to death (*Foster and Baron, 1996*).

The initial and required step in host cell intoxication is recognition of cell surface receptors by CT. In the 1970s, the ganglioside GM1 was identified as a host cell receptor for CT. A role for gangliosides was first postulated when Van Heyningen *et al.* discovered that a lipid extract from the brain inhibited CT activity (*van Heyningen et al., 1971*); subsequently, multiple groups showed that purified gangliosides inhibited CT binding, with GM1 the most potent inhibitor (*Cuatrecasas, 1973*; *Holmgren et al., 1973*; *King and van Heyningen, 1973*). To test whether GM1 could function as a receptor, exogenous GM1 was incorporated into host cell membranes, where it was shown to increase sensitivity to toxin, (*Cuatrecasas, 1973*) even sensitizing toxin-resistant cells (*Moss et al., 1976*). Holmgren and co-workers examined intestinal mucosa from several species and found that the extent of CT binding correlated with GM1 content (*Holmgren et al., 1975*). Further, addition of exogenous GM1 to intestinal mucosa resulted in increased secretory activity in response to CT stimulation, implying that GM1 serves as a functional receptor. Recognition of GM1 occurs exclusively through the CTB subunit. Indeed, the high affinity CTB-GM1 interaction has been extensively characterized through binding assays (*Kuziemko et al., 1996*) and x-ray crystallography analysis (*Merritt et al., 1994*).

CTB is closely related to the B subunit of *E. coli* heat-labile toxin (LTB) at the levels of sequence, (*Dallas and Falkow, 1980*) structure, (*Sixma et al., 1991*) and function (*Spangler, 1992*). While LTB is known to bind both GM1 and glycoprotein receptors, GM1 is commonly described to be the sole host cell receptor recognized by CTB (*Foster and Baron, 1996*). However, a variety of experimental approaches have pointed to the possibility that CTB may also recognize glycoproteins present on mammalian cells (*Morita et al., 1980*; *Monferran et al., 1990*; *Balanzino et al., 1994*; *Platt et al., 1997*; *Hansen et al., 2005*; *Blank et al., 2007*; *Day et al., 2012*). Indeed, CTB binding to cells does not uniformly parallel GM1 levels, implying the existence

of additional CTB-binding molecules (*Platt et al., 1997*; *Yanagisawa, 2006*). Moreover, GM1 binding does not always correlate with intoxication. For example, treatment of intestinal mucosa with *V. cholerae* sialidase yielded more GM1 but had no effect on toxin sensitivity (*Holmgren et al., 1975*). Also, a point mutant of CTB (H57A) was shown to maintain GM1 binding but the corresponding holotoxin did not intoxicate host cells (*Aman et al., 2001*). Finally, a recent analysis of a normal human intestinal epithelia found that GM1 comprises only 0.01% of the glycosphingolipid content, raising the question of whether its concentration in enterocytes is sufficient to account for intoxication by CT (*Breimer et al., 2012*).

Here we report that fucosylated molecules and glycoproteins are recognized by CTB and can function as receptors in host cell intoxication. Glycoproteins, not gangliosides, are responsible for the majority of CTB binding to human colonic epithelial cell lines. Using a metabolically incorporated photocrosslinking sugar analog, we isolated and identified one CTB-interacting glycoprotein, CEACAM5. The carbohydrate portion of the glycoprotein, not the amino acids, provides the CTB interaction motif. We show that fucose-containing glycans are recognized by CTB, resulting in internalization, the first step in host cell intoxication. Finally, we report evidence suggesting that fucosylated glycoconjugates recognized by CTB are present in normal human tissue. These results shed new light on mechanisms by which CT can enter and intoxicate host cells. In addition, the demonstration that CTB recognizes molecules other than GM1 has important implications for the interpretation of experiments where CTB is used to study the organization of lipids in the plasma membrane. Overall, these observations reveal a previously unrecognized, and potentially physiologically relevant, molecular mechanism for CT entry into epithelial cells.

## Results

### GM1 is not required for CTB crosslinking in human colonic epithelial cell lines

Previously, we reported a cell-permeable precursor sugar (Ac$_4$ManNDAz) that can be metabolized to a photocrosslinking sialic acid analog (SiaDAz) and incorporated into glycoconjugates – both glycoproteins and glycolipids – in place of natural sialic acids (*Figure 1A*) (*Tanaka and Kohler, 2008*). Culturing Jurkat cells, a human T cell line, with Ac$_4$ManNDAz results in production of SiaDAz-modified gangliosides, including SiaDAz-modified GM1 (*Bond et al., 2010*). After adding CTB to these SiaDAz-producing cells and applying UV radiation, we observed crosslinking of CTB to GM1, consistent with the idea that GM1 is the CT receptor (*Bond et al., 2010*, *2011*). Here we repeated the CTB crosslinking experiment in Jurkat cells, and also in two colonic epithelial cell lines, T84 and Colo205. We chose T84 cells because they are commonly used in studies of host cell intoxication by CT, (*Lencer, 1992*) and Colo205 cells as a second model of human colonic epithelia. By anti-CTB immunoblot analysis of Jurkat cell lysates, we confirmed detection of a CTB-containing species whose apparent mass (~13 kDa) matches the molecular weight of a CTB-GM1 complex (*Figure 1B*). In contrast, the CTB-GM1 complex was absent in lysates from both colonic epithelial cell lines (*Figure 1B*).

By reanalyzing the crosslinked lysates using a lower percent gel, we discovered additional CTB-reactive bands at much higher molecular weights in lysates from all three cell lines (Jurkat, T84 and Colo205; *Figure 1C*). Appearance of the high molecular weight bands was dependent on both inclusion of Ac$_4$ManNDAz and UV irradiation, suggesting that these bands also represent CTB crosslinked to sialylated molecules, but of much larger molecular weight than GM1. Surprised by the difference we observed between Jurkat and colonic epithelial cell lines, we also examined SiaDAz-mediated CTB crosslinking in two additional cell types, a human brain capillary endothelial cell line (hCMEC/D3) (*Weksler, 2005*) and a human bronchial epithelial cell line (HBEC) (*Ramirez, 2004*). In hCMEC/D3 cells, we observed both a complex with molecular weight consistent with CTB crosslinked to GM1, as well as a high molecular weight complex (*Figure 1—figure supplement 1A*). In HBECs, we also observed a complex consistent with CTB-GM1 crosslinking, as well as a faint higher molecular complex (*Figure 1—figure supplement 1B*). Thus, CTB crosslinking patterns are cell type dependent, and CTB crosslinking to GM1 is not observed in all cell types.

We considered two possible explanations for the high molecular weight CTB crosslinked complex. One possibility was that the complex represented CTB crosslinked to GM1, or another

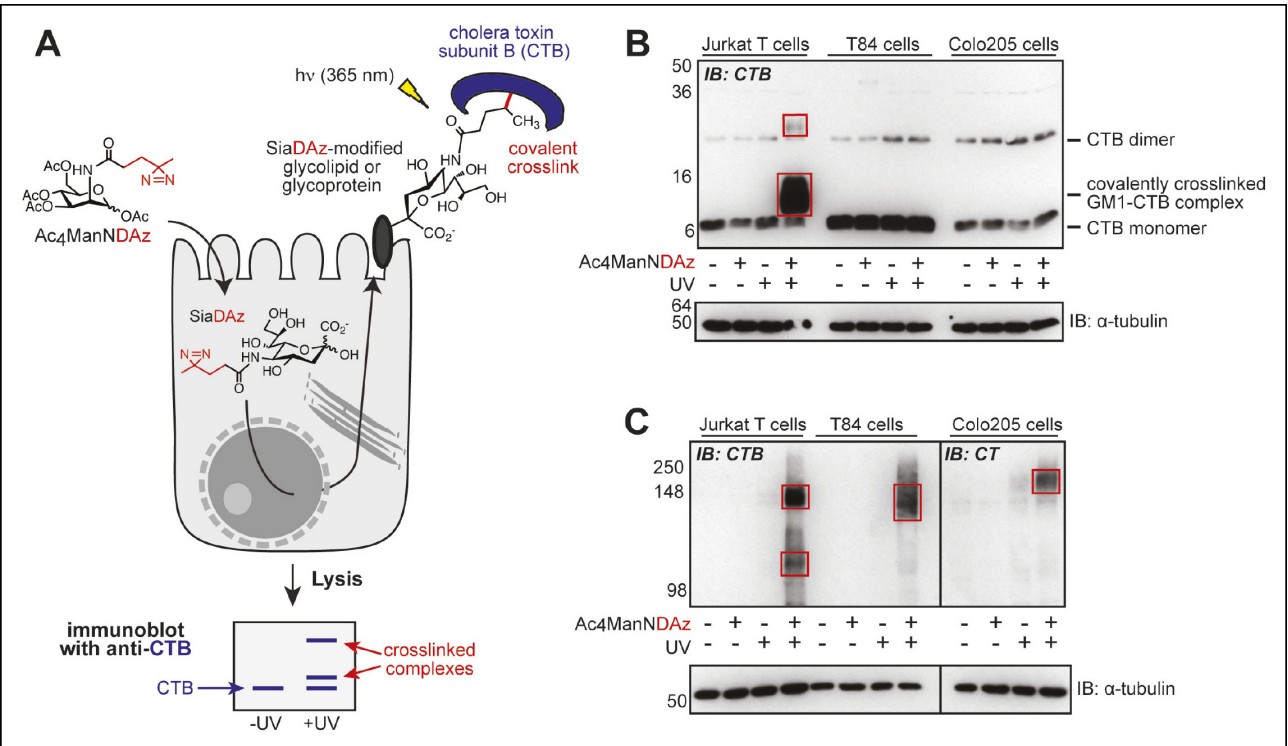

**Figure 1.** Products of SiaDAz-mediated crosslinking of CTB depend on cell type. (**A**) Photocrosslinking sialic acid (SiaDAz) is produced by culturing cells with Ac4ManNDAz. SiaDAz is incorporated into glycolipids and glycoproteins that are displayed on the cell surface. CTB is added to cells. Application of 365 nm radiation causes activation of the diazirine crosslinker and results in covalent crosslinking between CTB and neighboring SiaDAz-modified glycoconjugates. Crosslinked complexes can be observed by immunoblot, or purified and characterized by LC-MS/MS analysis. (**B**) Jurkat, T84, and Colo205 cells were cultured with Ac4ManNDAz, incubated with CTB, and UV irradiated. Lysates were analyzed by 15% SDS-PAGE immunoblot with anti-CTB antibody. Red boxes highlight crosslinked complexes not present in control lanes. (**C**) Jurkat, T84, and Colo205 cells were cultured with Ac4ManNDAz, incubated with CTB, and UV irradiated. Lysates were analyzed by 6% SDS-PAGE immunoblot with anti-CTB antibody (for Jurkat and T84 samples) or anti-CT antibody (for Colo205 samples). Red boxes highlight crosslinked complexes not present in control lanes.

The following figure supplements are available for Figure 1:

**Figure supplement 1.** SiaDAz-mediated crosslinking of CTB in additional cell types.

glycolipid, but behaving as an aggregate in the SDS-PAGE analysis. The second possibility was that the complex represented CTB crosslinked to a sialylated glycoprotein. To distinguish between these possibilities and to determine which class of glycoconjugates was required for formation of the high molecular weight complex, we made use of small molecule inhibitors and a decoy substrate that specifically interfere with production of different classes of glycoconjugates. The colonic epithelial T84 (*Figure 2A,B*) or Colo205 (*Figure 2C*) cell lines were therefore cultured with both Ac4ManNDAz and an inhibitor of glycosylation. CTB was added to the cells and crosslinking was performed. Lysates were examined by immunoblot using an anti-CT or anti-CTB antibody, and the effect of the various inhibitors on the intensity and molecular weight of the crosslinked band was determined.

The first inhibitor of glycosylation we used was NB-DGJ, a compound that interferes with glucosylation of ceramide, an early step in ganglioside biosynthesis (*Andersson et al., 2000*). We found that the CTB complexes detected by immunoblot in T84 and Colo205 cells were unaffected by culturing the cells with NB-DGJ (*Figure 2A,C*). In contrast, culturing Jurkat cells with NB-DGJ completely eliminates formation of the CTB-GM1 crosslinked complex (*Bond et al., 2010*). These data imply that the high molecular weight band does not represent an aggregate of the CTB-GM1 crosslinked complex.

Next, we examined inhibitors of protein glycosylation. To test if *N*-linked protein glycosylation is required for CTB crosslinking, cells were cultured with either deoxymannojirimycin or kifunensine, small molecules that interfere with the maturation of *N*-linked glycans (*Fuhrmann et al., 1984*;

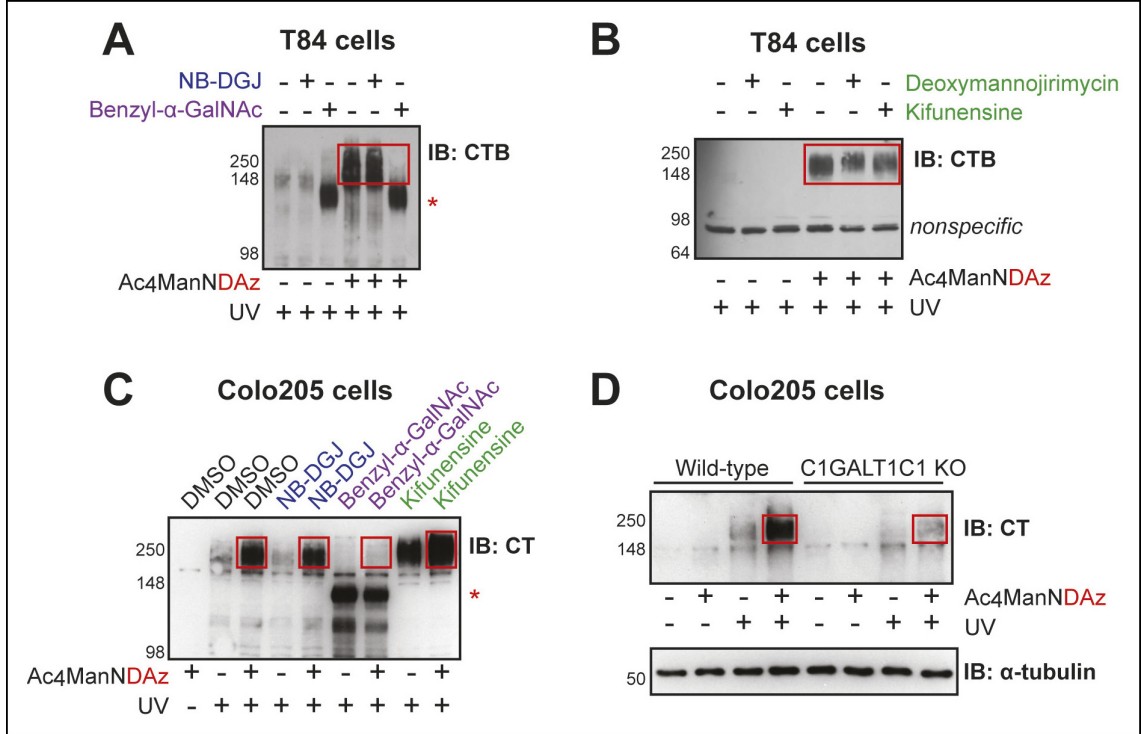

**Figure 2.** CTB recognizes glycoproteins on human colonic epithelial cell lines. (**A**) T84 cells were cultured with $Ac_4ManNDAz$ and a glycosylation inhibitor, incubated with CTB, and UV irradiated. NB-DGJ interferes with ganglioside biosynthesis; benzyl-$\alpha$-GalNAc competitively inhibits GalNAc-type *O*-linked glycosylation. Lysates were analyzed by 7.5% SDS-PAGE immunoblot with anti-CTB antibody. The asterisk indicates a SiaDAz-independent band that is observed with benzyl-$\alpha$-GalNAc treatment. (**B**) T84 cells were cultured with $Ac_4ManNDAz$ and a glycosylation inhibitor, incubated with CTB, and UV irradiated. Deoxymannojirimycin and kifunensine both interfere with maturation of *N*-linked glycans. Lysates were analyzed by 7.5% SDS-PAGE immunoblot with anti-CTB antibody. (**C**) Colo205 cells were cultured with $Ac_4ManNDAz$ and a glycosylation inhibitor, incubated with CTB, and UV irradiated. Lysates were analyzed by 6% SDS-PAGE immunoblot with anti-CT antibody. The red asterisk indicates a SiaDAz-independent band that is observed with benzyl-$\alpha$-GalNAc treatment. (**D**) Wild-type or C1GALT1C1 KO Colo205 cells were cultured with $Ac_4ManNDAz$, incubated with CTB, and UV irradiated. Lysates were analyzed by 6% SDS-PAGE immunoblot with anti-CT antibody. In all panels, red boxes highlight CTB crosslinked complexes observed in cells cultured with $Ac_4ManNDAz$ and treated with UV radiation.

The following figure supplements are available for Figure 2:

**Figure supplement 1.** Effectiveness of *N*-linked glycosylation inhibitors in human colonic epithelial cell lines.

**Figure supplement 2.** Effectiveness of *O*-linked glycosylation inhibitor in human colonic epithelial cell lines.

*Elbein et al., 1990*). Immature *N*-linked glycans will not contain SiaDAz and will be unable to engage in crosslinking. If the CTB crosslinked complex depends on *N*-linked protein glycosylation, then culturing cells with these inhibitors should reduce or eliminate the complex. We first confirmed the effectiveness of these inhibitors in T84 cells by observing a reduction in the apparent molecular weight of LAMP1, a protein with multiple sites of *N*-linked glycosylation (*Figure 2—figure supplement 1A*). Likewise, in Colo205 cells, we observed increased binding of lectin concanavalin A (ConA), reflecting accumulation of immature high mannose structures (*Figure 2—figure supplement 1B*). Both deoxymannojirimycin and kifunensine caused subtle effects on the high molecular weight crosslinked CTB complexes, altering the intensities and increasing the apparent molecular weight (*Figure 2B,C*). These results suggested that proteins with *N*-linked glycosylation play a role in the formation of crosslinked CTB complexes, but that *N*-linked glycosylation is not the sole contributor.

To test if *O*-linked protein glycosylation is required for CTB crosslinking, cells were cultured with benzyl-$\alpha$-GalNAc, a decoy substrate that competitively inhibits GalNAc-type *O*-linked glycosylation (*Kuan et al., 1989*). We first confirmed the effectiveness of benzyl-$\alpha$-GalNAc in T84 cells by showing that it caused a reduction in the apparent molecular weight of CD44, a protein with multiple sites of

O-linked glycosylation (*Figure 2—figure supplement 2A*). Benzyl-α-GalNAc also inhibited maturation of O-linked glycans in Colo205 cells, demonstrated by the observed increase in binding of the lectin peanut agglutinin (PNA), which binds to the T-antigen disaccharide (*Figure 2—figure supplement 2B*). Culturing either T84 or Colo205 cells with benzyl-α-GalNAc resulted in dramatic reductions in the intensity of the high molecular weight crosslinked complexes, suggesting that CTB crosslinks to glycoproteins bearing O-linked glycans (*Figure 2A,C*). However, use of benzyl-α-Gal-NAc also resulted in the appearance of a new CTB-containing species at lower apparent molecular weight (~100 kDa). Because appearance of the 100 kDa band was not dependent on the addition of $Ac_4ManNDAz$, it does not represent crosslinking through SiaDAz, and may relate to the UV absorbance properties of the benzyl group in benzyl-α-GalNAc. Overall, the results of the inhibition experiments demonstrate that CTB crosslinks to glycoproteins, with both N-linked and O-linked glycans playing roles.

Because of the potential for small molecule inhibitors to exert unanticipated effects, we sought a second approach to gain insight into the role of glycoproteins in CTB crosslinking. We used Colo205 cells in which elongation of GalNAc-type O-linked glycans is blocked due to zinc finger nuclease (ZFN) targeting of *C1GALT1C1*, which encodes a chaperone required for biosynthesis of GalNAc-type O-linked glycans (*Steentoft et al., 2011*). While wild-type Colo205 cells produced the high molecular weight crosslinked CTB complex, the intensity of this band was dramatically reduced in cells lacking C1GALT1C1 activity (*Figure 2D*). We conclude that GalNAc-type O-linked glycosylation of proteins plays an important role in formation of high molecular weight crosslinked CTB complexes in Colo205 cells. Taken together, the crosslinking data indicate that CTB crosslinks to glycoproteins in colonic epithelial cells, with both N-linked and O-linked glycosylation of proteins making contributions. In contrast, no evidence pointed to CTB crosslinking to GM1 or other gangliosides in either colonic epithelial cell line.

## Colonic epithelial cell lines contain little or no GM1 ganglioside

Our inability to detect CTB crosslinking to GM1 in either colonic epithelial cell line stimulated us to evaluate the ganglioside content of these cells. Using Soxhelt extraction, lipids were extracted from T84, Colo205, and Jurkat cells cultured with either vehicle or the glycosphingolipid inhibitor NB-DGJ. Alkali-labile phospholipids were removed by mild alkaline hydrolysis, and some non-polar compounds were removed by silicic acid chromatography. When analyzed by thin-layer chromatography and resorcinol staining (*Figure 3A*), the fraction from untreated Jurkat cells yielded several bands migrating with mobilities similar to and slower than the GM3 ganglioside (*Figure 3A*, lane 5). In contrast, in the fraction from Jurkat cells cultured with NB-DGJ, only the most slow-migrating band was present (*Figure 3A*, lane 6). The fraction from T84 cells produced two bands, one migrating between GM3 and GM1 gangliosides, and one migrating similarly to GM1 (*Figure 3A*, lane 1). Finally, the fraction from Colo205 cells was only faintly stained (*Figure 3A*, lane 3).

To further identify these species, the partially purified glycosphingolipid fractions were probed for binding to $^{125}$I-labeled CTB, also in thin-layer chromatography format. CTB bound strongly to the fraction isolated from untreated Jurkat cells (*Figure 3B*, lane 5), and also more weakly to the fraction isolated from Jurkat cells cultured with NB-DGJ (*Figure 3B*, lane 6). The CTB-binding material from Jurkat cells co-migrated with GM1, and appeared as a doublet, likely corresponding to GM1 species with different ceramide components. No binding of $^{125}$I-labeled CTB to crude glycosphingolipid fractions from T84 or Colo205 cells was observed (*Figure 3B*, lanes 1 and 3), not even when high concentrations of the glycosphingolipid fractions and high concentrations of $^{125}$I-CTB were used. Based on the amount of cells analyzed and the sensitivity of detection, we estimate that T84 and Colo205 cells contain no more than 5 ng of GM1 per million cells.

To evaluate the effect of NB-DGJ on glycosphingolipid production, binding of the Galβ4GlcNAc-binding lectin from *E. cristagalli* (*Teneberg et al., 1994*) to the partially purified glycosphingolipid fractions was tested. For the crude glycosphingolipid fraction from T84 cells, binding in the tetraglycosylceramide region, most likely to neolactotetraosylceramide (Galβ4GlcNAcβ3Galβ4Glcβ1Cer), was apparent (*Figure 3C*, lane 1), but this binding was not visible in the fraction from T84 cells cultured with NB-DGJ (*Figure 3C*, lane 2). We conclude that NB-DGJ effectively inhibits production of glucosylceramide glycolipids in T84 cells. Thus, if a low, undetectable amount of GM1 is present in the intestinal epithelial cell lines, it is reasonable to assume that this level is further reduced by culturing the cells with NB-DGJ.

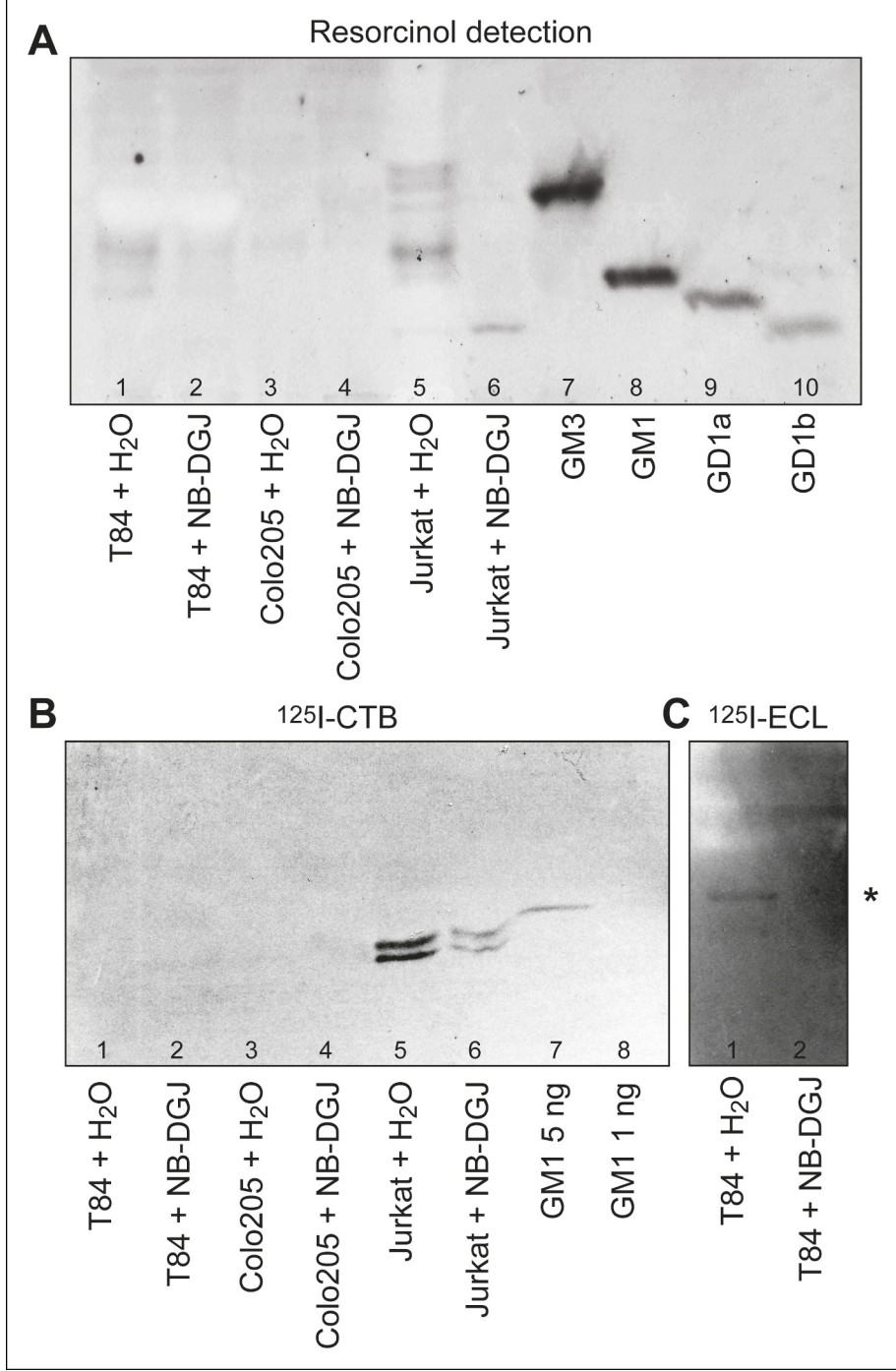

**Figure 3.** HP-TLC analysis of glycosphingolipids from T84, Colo205, and Jurkat cells. Partially purified glycosphingolipid fractions isolated from T84, Colo205 and Jurkat cells cultured with either vehicle or the glycosphingolipid inhibitor NB-DGJ were separated on aluminum-backed silica gel plates using chloroform/methanol/water (60:35:8, by volume) as solvent and stained with resorcinol (A). Chromatograms with separated glycosphingolipids were incubated with $^{125}$I-labeled CTB (B) or lectin from *Erythrina cristagalli* (C), followed by autoradiography for 12 hr. In (C), the asterisk (*) highlights the putative neolactotetraosylceramide band. Additional detail about samples analyzed is provided in the methods section.

## Glycoproteins are the dominant CTB-binding molecules on the surface of human colonic epithelial cell lines

With the insight that CTB can recognize glycoproteins displayed on the colonic epithelial cells, we next assessed the relative contributions of different glycoconjugates to overall CTB binding in different cell types. First we examined Jurkat cells, where we had observed multiple CTB crosslinked species. Jurkat cells were cultured with inhibitors of glycosylation to prevent production of specific classes of glycoconjugates, then CTB binding was measured by flow cytometry (*Figure 4A*). The ganglioside biosynthesis inhibitor NB-DGJ reduced GM1 production in Jurkat cells (*Figure 3B*) and also resulted in a decrease in CTB binding (*Figure 4A*). In contrast, culturing Jurkat cells with benzyl-α-GalNAc or kifunensine had no significant effect on CTB binding (*Figure 4A*). The inability of benzyl-α-GalNAc to affect CTB binding to Jurkat cells is consistent with the known *O*-linked glycosylation defect in these cells, caused by a frame-shift mutation in the gene encoding chaperone C1GALT1C1 (*Figure 4—figure supplement 1A*) (*Ju and Cummings, 2002*). Jurkat cells do produce *N*-linked

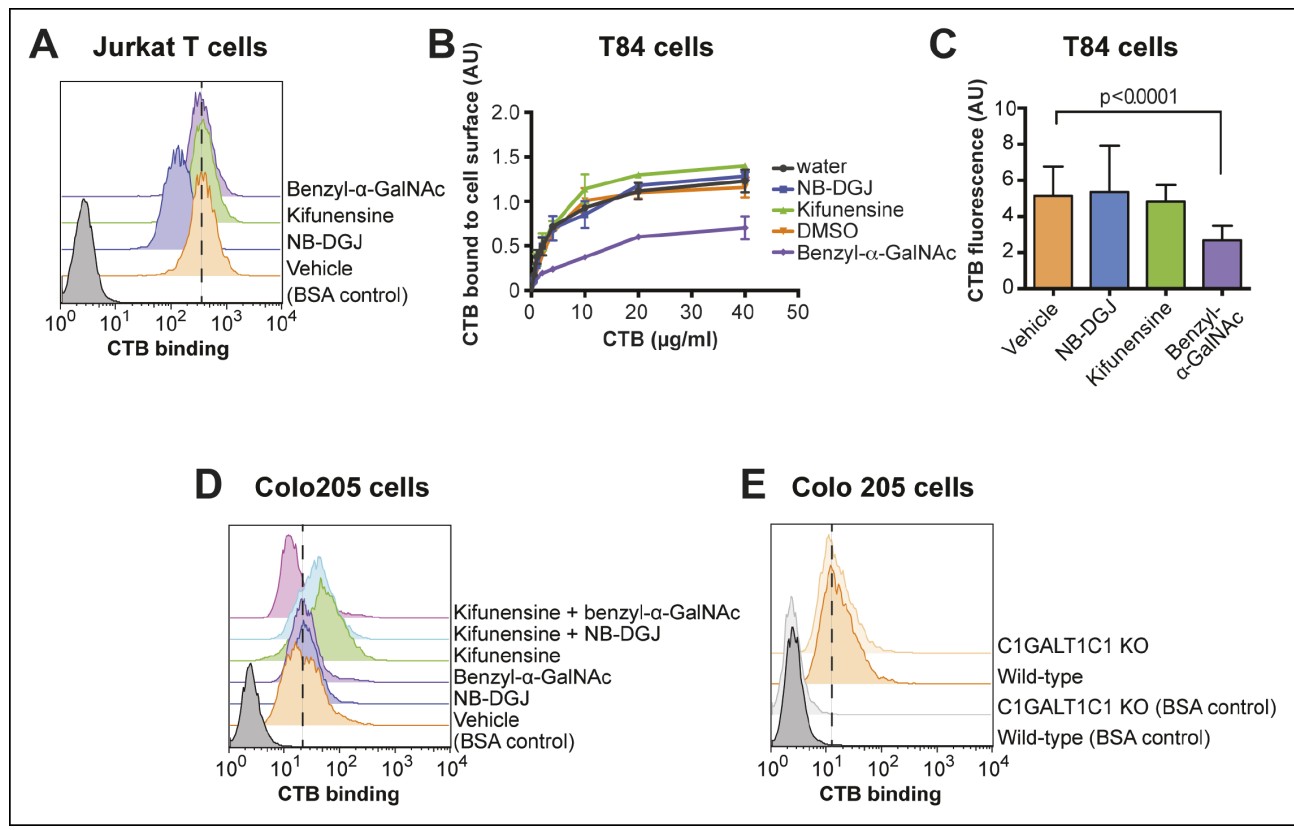

**Figure 4.** CTB binding to human colonic epithelial cell lines depends on protein glycosylation. (**A**) Jurkat cells were cultured with inhibitors of glycosylation. Binding of CTB was measured by flow cytometry. Data shown are a single representative trial from two independent experiments. (**B**) T84 cells were cultured with inhibitors of glycosylation, then incubated with increasing concentrations of CTB. Binding of CTB was measured by ELISA. Data presented are the mean values for duplicate samples with error bars indicating the standard deviation. A replicate experiment yielded similar results. (**C**) T84 cells were cultured with inhibitors of glycosylation. Binding of Alexa Fluor 647-CTB was measured by fluorescence microscopy. (**D**) Colo205 cells were cultured with inhibitors of glycosylation. Binding of CTB was measured by flow cytometry. Data shown are a single representative trial from two independent experiments. (**E**) Binding of CTB to wild-type or C1GALT1C1 KO Colo205 cells was measured by flow cytometry. Data shown are a single representative trial from three independent experiments.

The following figure supplements are available for Figure 4:

**Figure supplement 1.** Characterization of protein glycosylation in Jurkat T cells.

**Figure supplement 2.** Representative fluorescence microscopy images of Alexa Fluor 647-CTB binding to T84 cells cultured with glycosylation inhibitors.

glycans, as evidenced by the enhancement of ConA binding to kifunensine-treated cells (*Figure 4—figure supplement 1B*), but since kifunensine treatment does not affect CTB binding to Jurkat cells (*Figure 4A*), *N*-linked glycans do not appear to be major contributors to CTB binding. Thus, the inhibition studies indicate that gangliosides are the dominant binding partners for CTB on Jurkat cells.

Next, we turned attention to the colonic epithelial cell lines. Because of difficulties associated with performing flow cytometry on the highly adherent T84 cells, CTB binding to the surface of T84 cells was measured by other methods. In the first approach, binding of biotin-labeled CTB to T84 monolayers was measured by an ELISA method (*Figure 4B*). In the second approach, binding of Alexa Fluor 647-labeled CTB to clusters of cells was quantified by fluorescence microscopy (*Figure 4C* and *Figure 4—figure supplement 2*). In the fluorescence microscopy approach, fluorescence was observed primarily to the outer surface of cell clusters, suggesting that Alexa Fluor 647-labeled CTB has limited access to the interior of the cell clusters, or that the CTB ligand is not displayed on the interior surface. Despite the differences in format, the results obtained by the ELISA and fluorescence microscopy methods were in agreement. NB-DGJ, the inhibitor of ganglioside biosynthesis, had no effect on CTB binding. Kifunensine, an inhibitor of *N*-linked glycan maturation, resulted in only small effects on CTB binding. In contrast, culturing cells with benzyl-α-GalNAc, which blocks elaboration of *O*-linked glycans, resulted in reductions in CTB binding in both assays. We conclude the glycoproteins are the dominant CTB binding partners in T84 cells and that gangliosides do not make a large contribution to CTB binding to T84 cells.

To measure CTB binding to Colo205 cells, we used flow cytometry. NB-DGJ treatment did not affect CTB binding to Colo205 cells (*Figure 4D*). However, in contrast to the results observed for T84 cells, benzyl-α-GalNAc did not cause a reduction in CTB binding to Colo205 cells (*Figure 4D*), nor did Colo205 cells lacking C1GALT1C1 show reduced CTB binding (*Figure 4E*). Instead, Colo205 cells cultured with the *N*-linked inhibitor kifunensine exhibited enhanced CTB binding (*Figure 4D*), consistent with a small increase in SiaDAz crosslinking that kifunensine causes in these cells (*Figure 2C*). We considered the possibility that CTB binds to multiple classes of glycoconjugates in Colo205 cells. We cultured Colo205 cells with pairs of inhibitors and measured CTB binding. The only case where we observed decreased CTB binding was when the two protein glycosylation inhibitors – kifunensine and benzyl-α-GalNAc – were used together (*Figure 4D*). In no case did culturing cells with NB-DGJ result in decreased CTB binding. Based on these data, we propose that CTB binds primarily to glycoproteins on Colo205 cells, with contributions from both *N*-linked and *O*-linked glycans. Gangliosides do not make a large contribution to CTB binding to Colo205 cells.

## CTB interacts directly with glycoproteins, including CEACAM5

To demonstrate conclusively that CTB binds glycoproteins, we isolated crosslinked CTB complexes and used mass spectrometry to identify one of the crosslinked glycoproteins. T84 cells were cultured with Ac₄ManNDAz, and then incubated with biotin-CTB. After UV irradiation, the cells were lysed and the membrane fraction isolated. Biotinylated, crosslinked complexes were purified on streptavidin-agarose and subjected to trypsin digest, followed by LC-MS/MS analysis. Alternatively, purified crosslinked material was loaded onto an SDS-PAGE gel. Then the CTB-glycoprotein crosslinked material was excised, trypsin digest was performed, and the released peptides were analyzed by LC-MS/MS. We focused attention on proteins that were detected with a spectral count higher than 3 in either crosslinked sample, and not detected in the corresponding control sample (*Table 1*).

CEACAM5 (also known as CEA) and CD44 were among the top hits from the proteomics analysis and are both known to be highly glycosylated. We tested whether either of these proteins were present in the CTB-glycoprotein crosslinked complex. Cells were cultured with Ac₄ManNDAz, then incubated with CTB. After UV irradiation, the cells were lysed. Immunoprecipitation of CEACAM5 resulted in purification of the CTB crosslinked complex (*Figure 5A*). Similarly, immunoprecipitation performed with anti-CD44 resulted in purification of the CTB crosslinked complex (*Figure 5B*). We also performed immunoprecipitation with anti-CTB, which resulted in efficient purification of CEACAM5, but only weak purification of CD44 (*Figure 5C*). Notably, anti-CTB immunoprecipitated CEACAM5 even when the Ac₄ManNDAz crosslinker was omitted, suggesting that the CTB-CEACAM5 interaction is relatively strong (*Figure 5C,D*). Furthermore, the material purified by anti-CTB and recognized by anti-CEACAM5 had a slightly higher apparent molecular weight when the crosslinker was included than when it was omitted (*Figure 5C*). Thus, we observe direct crosslinking between

**Table 1.** Proteomics analyses of CTB crosslinked complexes.

| Protein symbol | Protein name | In gel | | | In solution | | |
|---|---|---|---|---|---|---|---|
| | | Peptide sequences | % coverage | Spectral count | Peptide sequences | % coverage | Spectral count |
| ITGB4 | Integrin beta-4 | 11 | 6.5 | 14 | NA | NA | NA |
| SLC12A2 | Solute carrier family 12 (Sodium/potassium/ chloride transporters), member 2 (isoform CRA) | 6 | 6.7 | 10 | NA | NA | NA |
| CD44 | CD44 antigen | 5 | 28.6 | 7.92 | NA | NA | NA |
| PLXNB2 | Plexin-B2 | 4 | 3 | 6 | 5 | 3.5 | 3.00 |
| CEACAM5 | Carcinoembryonic antigen-related cell adhesion molecule 5 | 3 | 6.3 | 5 | 2 | 3.2 | 4.00 |
| CTB | Cholera toxin subunit B | 2 | 15.3 | 4 | 8 | 53.2 | 113.00 |
| LY75 | Lymphocyte antigen 75 (isoform 4) | 3 | 2.1 | 3 | 9 | 6.3 | 11.00 |
| COPA | Coatomer subunit alpha | 3 | 3 | 3 | NA | NA | NA |
| SPTB2 | Spectrin beta chain, brain 1 | 4 | 1.8 | 3 | NA | NA | NA |
| ITGA6 | Integrin alpha-6 (isoform Alpha-6X1A) | NA | NA | NA | 13 | 12.7 | 14.83 |
| EGFR | Epidermal growth factor receptor (isoform 1) | NA | NA | NA | 12 | 11.5 | 11.92 |
| MUC13 | Mucin-13 | NA | NA | NA | 6 | 15.6 | 9.00 |
| ITGB1 | Integrin beta-1 (isoform Beta-1A) | NA | NA | NA | 7 | 9.3 | 7.96 |
| DPP4 | Dipeptidyl peptidase 4 | NA | NA | NA | 8 | 10.1 | 7.00 |
| CDCP1 | CUB domain-containing protein 1 (isoform 1) | NA | NA | NA | 6 | 6.3 | 6.99 |
| PLXNA1 | Plexin-A1 | NA | NA | NA | 8 | 4.2 | 6.98 |
| SPINT1 | Kunitz-type protease inhibitor 1 (isoform 2) | NA | NA | NA | 6 | 11.3 | 6.00 |
| PARP4 | Poly [ADP-ribose] polymerase 4 | NA | NA | NA | 6 | 3.8 | 6.00 |
| ITGAV | Isoform 1 of Integrin alpha-V (isoform 1) | NA | NA | NA | 5 | 5.1 | 5.00 |
| ATP1B3 | Sodium/potassium-transporting ATPase subunit beta-3 | NA | NA | NA | 4 | 17.2 | 4.97 |
| PTGFRN | PTGFRN protein (Fragment) | NA | NA | NA | 6 | 10.6 | 4.00 |
| SCARB1 | Scavenger receptor class B member 1 (isoform 1) | NA | NA | NA | 4 | 9.3 | 4.00 |
| PGRMC1 | Membrane-associated progesterone receptor component 1 | NA | NA | NA | 3 | 15.4 | 4.00 |
| DSG2 | Desmoglein-2 | NA | NA | NA | 4 | 4.4 | 4.00 |
| PVR | Poliovirus receptor (isoform beta) | NA | NA | NA | 4 | 15.7 | 4.00 |
| COPB1 | Coatomer subunit beta | NA | NA | NA | 4 | 5.4 | 4.00 |
| PDIA4 | Protein disulfide-isomerase A4 | NA | NA | NA | 4 | 6.2 | 3.00 |
| ST14 | Suppressor of tumorigenicity 14 protein | NA | NA | NA | 3 | 4.9 | 3.00 |

*Table 1. continued on next page*

CTB and CEACAM5. Additionally, CD44 co-purifies with the CTB-glycoprotein crosslinked complex, but is not directly crosslinked to CTB.

We performed additional experiments to characterize the role of protein glycosylation in the CTB-CEACAM5 interaction. We cultured cells with inhibitors of glycosylation and assessed the effect on the noncovalent CTB-CEACAM5 interaction (*Figure 5D*). Both benzyl-α-GalNAc and kifunensine treatment caused reductions in the apparent molecular weight of CEACAM5, suggesting that this protein bears both *N*-linked and *O*-linked glycans. Treatment with benzyl-α-GalNAc enhanced the CTB-CEACAM5 interaction, while treatment with kifunensine abrogated the interaction, and treatment with NB-DGJ had no effect. Inhibition of *N*-linked glycan maturation also blocked crosslinking between CTB and CEACAM5 (*Figure 5E*). Thus, protein glycosylation regulates the ability of CEACAM5 to interact with CTB.

Taken together, these data demonstrate a direct interaction between CTB and at least one glycoprotein, CEACAM5. Recognition of CEACAM5 by CTB depends on protein glycosylation, suggesting that the glycan, and not the protein, may provide the recognition determinant. While culturing cells with kifunensine abolished the CEACAM5-CTB interaction (*Figure 5E*), overall CTB crosslinking in T84 cells was only slightly affected by kifunensine treatment (*Figure 2B*). Thus, additional CTB-crosslinking proteins exist in T84 cells and remain to be identified.

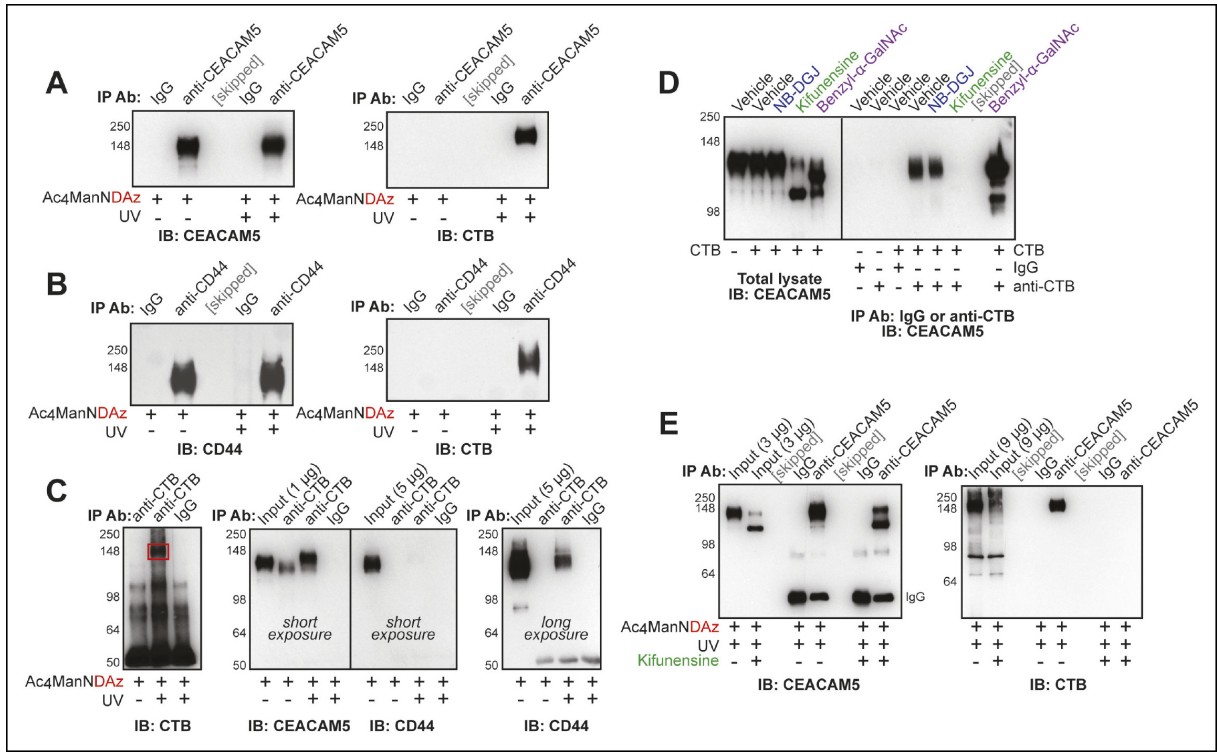

**Figure 5.** Protein glycosylation is required for a CTB-glycoprotein interaction. (**A**) T84 cells were cultured with Ac4ManNDAz, incubated with CTB, and UV irradiated. Immunopurification was performed with control IgG or anti-CEACAM5. 7.5% SDS-PAGE immunoblots were performed with anti-CEACAM5 and anti-CTB. (**B**) T84 cells were cultured with Ac4ManNDAz, incubated with CTB, and UV irradiated. Immunopurification was performed with control IgG or anti-CD44. 7.5% SDS-PAGE immunoblots were performed with anti-CD44 and anti-CTB. (**C**) T84 cells were cultured with Ac4ManNDAz, incubated with CTB, and UV irradiated. Immunopurification was performed with control IgG or anti-CTB. 6% SDS-PAGE immunoblots were performed with anti-CTB, anti-CEACAM5, and anti-CD44. (**D**) T84 cells were cultured with inhibitors of glycosylation, then incubated with or without CTB. Immunopurification was performed with control IgG or anti-CTB. 6% SDS-PAGE immunoblots were performed with anti-CEACAM5. (**E**) T84 cells were cultured with Ac4ManNDAz and with or without kifunensine, incubated with CTB, and UV irradiated. Immunopurification was performed with control IgG or anti-CEACAM5. 7.5% SDS-PAGE immunoblots were performed with anti-CEACAM5 and anti-CTB.

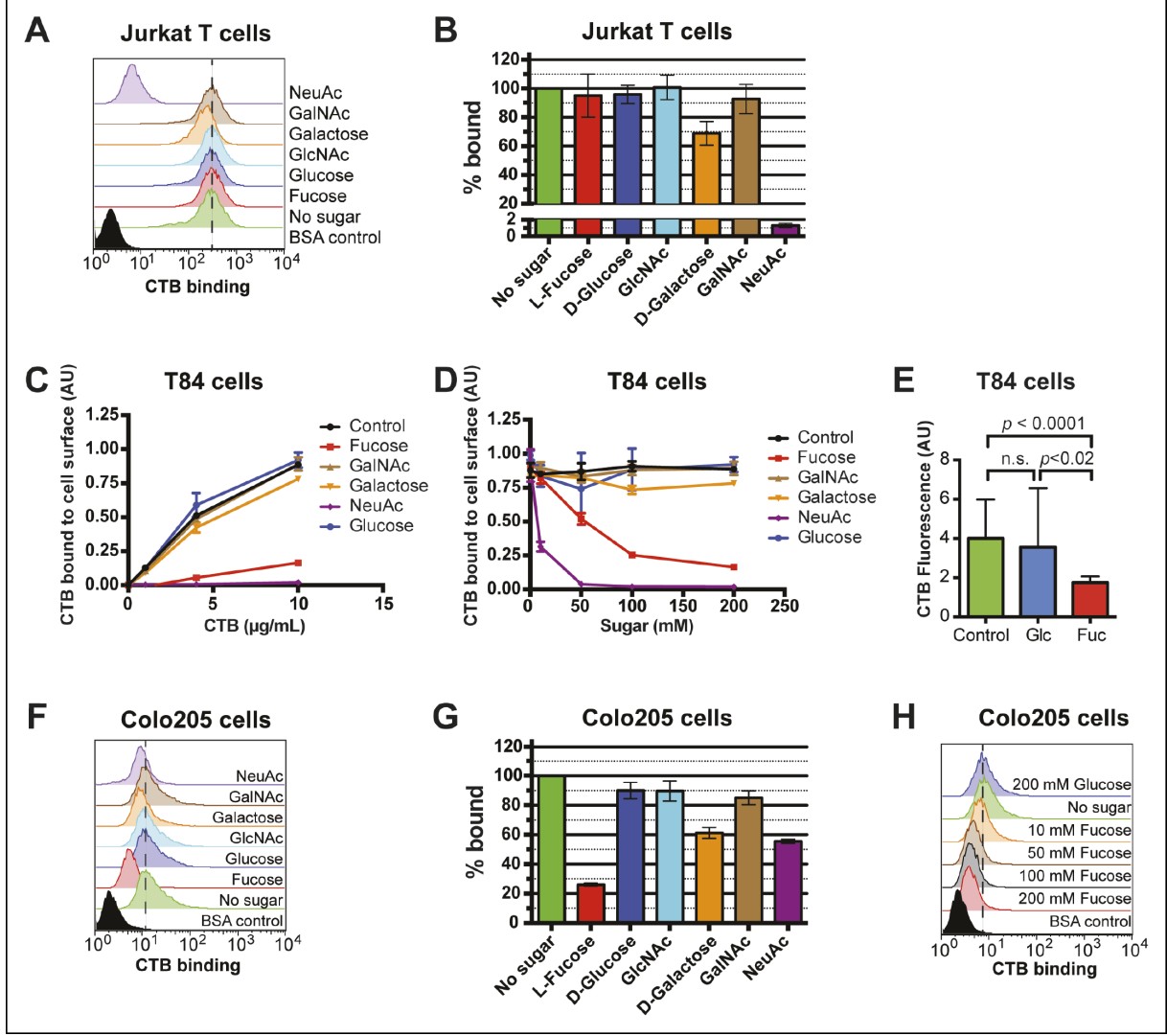

**Figure 6.** Fucose blocks binding of CTB to human colonic epithelial cell lines. (A) Jurkat cells were incubated with 4 µg/mL of CTB in the presence of 100 mM of free sugar. Binding of CTB was measured by flow cytometry. Data shown are a single representative trial from three independent experiments. (B) The median fluorescence intensity (MFI) for the no sugar treatment sample presented in panel A was normalized to 100% bound. Data shown represent an average of three independent trials and their standard deviations. (C) T84 cells were incubated with 200 mM of free sugar and variable concentrations of CTB. Binding of CTB was measured by ELISA. Data presented are the mean values for duplicate samples with error bars indicating the standard deviation. A replicate experiment yielded similar results. (D) T84 cells were incubated with variable free sugar concentrations and 10 µg/mL of CTB. Binding of CTB was measured by ELISA. Data presented are the mean values for duplicate samples with error bars indicating the standard deviation. A replicate experiment yielded similar results. (E) T84 cells were incubated with 100 mM fucose or 100 mM glucose in the presence of Alexa Fluor 647-CTB. Binding of Alexa Fluor 647-CTB was measured by fluorescence microscopy. (F) Colo205 cells were incubated with 10 µg/mL of CTB in the presence of 100 mM of free sugar. Binding of CTB was measured by flow cytometry. Data shown are a single representative trial from three independent experiments. (G) The median fluorescence intensity (MFI) for the no sugar treatment sample presented in panel F was normalized to 100% bound. Data shown represent an average of three independent trials and their standard deviations. (H) Colo205 cells were incubated with variable free fucose concentrations and 10 µg/mL of CTB. Binding of CTB was measured by flow cytometry. Data shown are a single representative trial from two independent experiments.

The following figure supplements are available for Figure 6:

**Figure supplement 1.** Effects of free sugars on lectin binding to Jurkat cells.

**Figure supplement 2.** Representative fluorescence microscopy images showing effects of free sugars on binding of Alexa Fluor 647-CTB to T84 cells.

**Figure supplement 3.** Effects of free sugars on lectin binding to Colo205 cells.

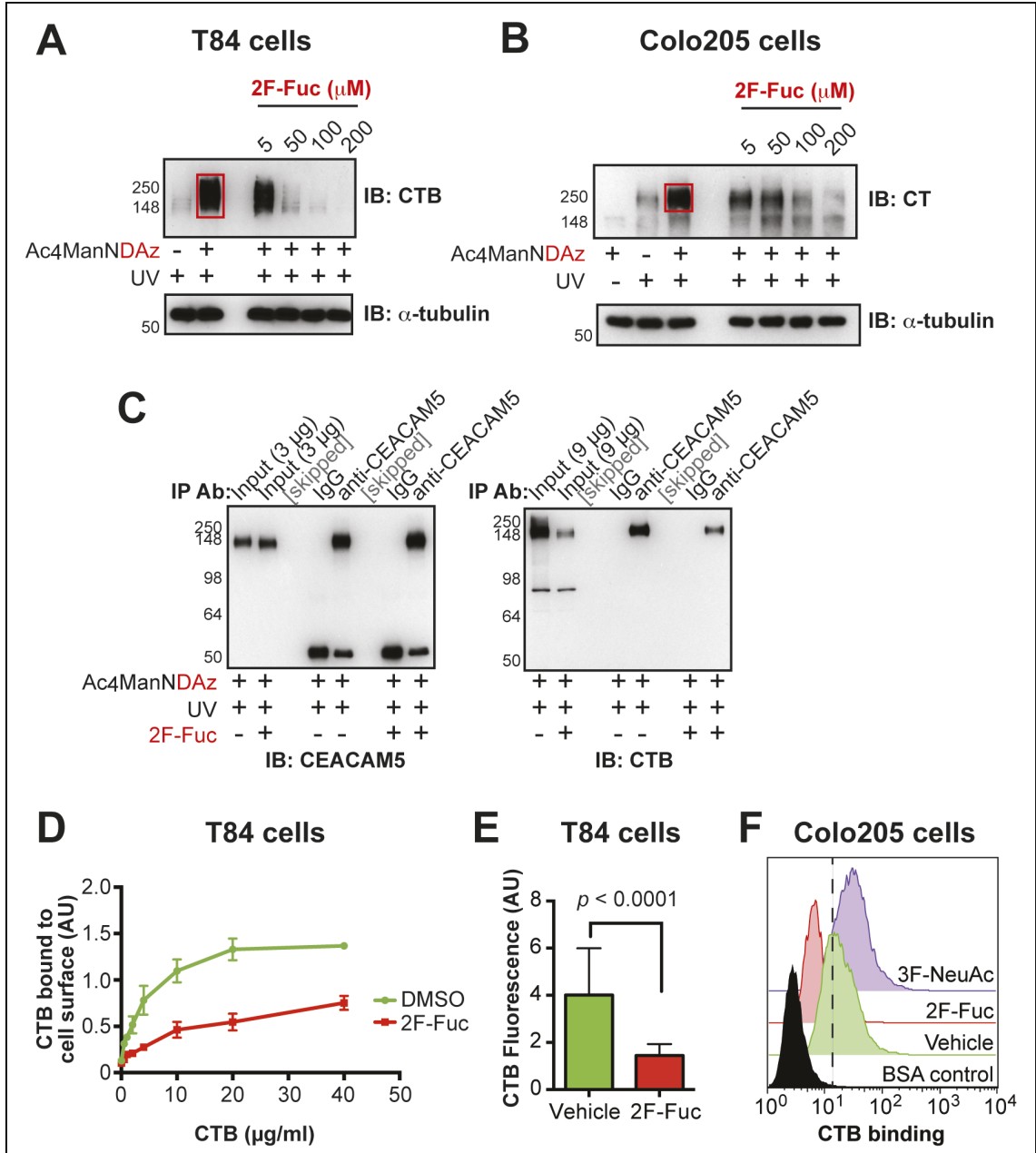

**Figure 8.** Inhibition of fucosylation reduces CTB crosslinking and binding to human colonic epithelial cell lines. (**A**)T84 cells were cultured with Ac₄ManNDAz and increasing concentrations of 2F-Fuc, incubated with CTB, and UV irradiated. Lysates were analyzed by 7.5% SDS-PAGE immunoblot with anti-CTB antibody. (**B**)Colo205 cells were cultured with Ac₄ManNDAz and increasing concentrations of 2F-Fuc, incubated with CTB, and UV irradiated. Lysates were analyzed by 7.5% SDS-PAGE immunoblot with anti-CT antibody. (**C**) T84 cells were cultured with Ac₄ManNDAz and with or without 2F-Fuc, incubated with CTB, and UV irradiated. Immunopurification was performed with IgG or anti-CEACAM5. 7.5% SDS-PAGE immunoblots were performed with anti-CEACAM5 and anti-CTB. (**D**)T84 cells were cultured with 2F-Fuc, then incubated with increasing concentrations of CTB. Binding of CTB was measured by ELISA. Data presented are the mean values for duplicate samples with error bars indicating the standard deviation. A replicate experiment yielded similar results. (**E**) T84 cells were cultured with 2F-Fuc. Binding of Alexa Fluor 647-CTB was measured by fluorescence microscopy. (**F**) Colo205 cells were cultured with 2F-Fuc or 3F-NeuAc. Binding of CTB was measured by flow cytometry. Data shown are a single representative trial from three independent experiments.

The following figure supplements are available for Figure 8:

**Figure supplement 1.** Culturing T84 cells with 2F-Fuc causes decreased cell surface fucosylation.

*Figure 8. Continued*

**Figure supplement 2.** Representative fluorescence microscopy images of Alexa Fluor 647-CTB binding to T84 cells cultured with 2F-Fuc.

**Figure supplement 3.** Culturing Colo205 cells with 2F-Fuc and 3F-NeuAc causes decreased cell surface fucosylation and sialylation, respectively.

# Fucose is important for CTB binding to human intestinal epithelial cell lines

Because glycan identity, not protein identity, appeared to be essential to CTB binding, our next goal was to gain greater insight into glycan structures recognized by CTB. Epidemiological studies implicate variation among fucose-containing glycoconjugates in human susceptibility to cholera (*Barua and Paguio, 1977*; *Swerdlow et al., 1994*; *Harris et al., 2005*; *Holmner et al., 2010*). In addition, some glycan array binding experiments performed by the Consortium for Functional Glycomics (CFG) suggest that fucosylated glycans can be recognized by CTB, albeit with lower avidity than GM1 (*Consortium for Functional Glycomics, 2010*). Importantly, fucose is not present in GM1, which is recognized by CTB through interactions with its terminal Neu5Ac and galactose residues (*Merritt et al., 1994*). We therefore tested whether monosaccharides, including fucose, Neu5Ac, and galactose, could competitively inhibit CTB binding to different cell lines. We found that 100 mM Neu5Ac potently blocked CTB binding to Jurkat cells, while 100 mM galactose inhibited slightly (*Figure 6A,B*). Other monosaccharides had no effect. In particular, 100 mM fucose did not interfere with CTB binding to Jurkat cells (*Figure 6B*), even though binding of a fucose-recognizing lectin, *Aleuria aurantia* lectin (AAL), was completely inhibited by the same concentration of free fucose (*Figure 6—figure supplement 1*). These results are consistent with CTB binding to Jurkat cells via recognition of GM1.

Competitive inhibition of CTB binding by monosaccharides was also assessed on colonic epithelial cell lines. T84 cells were incubated with increasing amounts of biotin-CTB in a buffer containing 200 mM free monosaccharide. Binding of biotin-CTB was measured by an ELISA method (*Figure 6C*). Fucose and Neu5Ac, but not other sugars, effectively prevented binding of all concentrations of CTB. Further, both fucose and Neu5Ac inhibited biotin-CTB binding to T84 cells in a concentration-dependent manner (*Figure 6D*). Free fucose also blocked binding of Alexa Fluor 647-CTB to T84 cells, as measured by fluorescence microscopy (*Figure 6E* and *Figure 6—figure supplement 2*). The ability of monosaccharides to inhibit binding of biotin-CTB to Colo205 cells was measured by flow cytometry (*Figure 6F*). Fucose was the most effective inhibitor, while galactose and Neu5Ac each showed moderate inhibition (*Figure 6G*). While as little as 10 mM fucose could interfere with biotin-CTB binding to Colo205 cells, glucose did not affect biotin-CTB binding even at concentrations as high as 200 mM (*Figure 6H*). Similarly, free fucose, but not free Neu5Ac or glucose, blocked binding of fucose-recognizing AAL to Colo205 cells (*Figure 6—figure supplement 3*) . A theme emerging from the monosaccharide competition experiments was that fucose specifically inhibits CTB binding to colonic epithelial cell lines, but not to Jurkat cells. In addition, Neu5Ac and galactose each displayed some ability to block binding of CTB to multiple cell lines.

Based on the results of the monosaccharide competition experiments, we used lectins as blocking reagents to assess whether fucosylated and/or sialylated structures are recognized by CTB. T84 cells were incubated with lectin, then variable concentrations of biotin-CTB were added and biotin-CTB binding was measured by ELISA (*Figure 7A*). Treatment with *Ulex europaeus* agglutinin I (UEA-1) or *Lotus tetragonolobus* lectin (LTL), each of which recognize specific fucosylated structures, had no effect on biotin-CTB binding. We also examined AAL and *Lens culinaris* agglutinin (LCA), which both recognize α1-6-fucosylated structures, although AAL has a broader substrate scope and also binds fucose in other linkages (*Kochibe and Furukawa, 1980*; *Matsumura et al., 2007*; *Yu et al., 2012*). AAL was able to block biotin-CTB binding, but LCA was not. We titrated the amount of lectins used in this blocking assay, while holding the biotin-CTB concentration constant (*Figure 7B*). AAL, but not other lectins, blocked biotin-CTB binding to T84 cells in a concentration-dependent way. We used flow cytometry to test the ability of lectins to block CTB binding to Colo205 cells. Colo205 cells were first incubated with lectin, then CTB was added. Similar to what we observed for T84 cells, AAL effectively blocked CTB binding, while other lectins had no effect (*Figure 7C*). AAL blocked CTB binding to Colo205 cells in a concentration-dependent way (*Figure 7D*), and when free fucose was included during the AAL pre-incubation, AAL was no longer able to block CTB binding (*Figure 7E*). While AAL effectively blocked CTB binding to both T84 and Colo205 cells, this fucose-recognizing lectin was not able to block biotin-CTB binding to Jurkat T cells, even when used in excess over toxin concentration (*Figure 7F*). Taken together, results from lectin blocking experiments are consistent with the idea that CTB binds fucosylated structures on the surface of colonic epithelial cell lines, while fucosylated structures do not contribute significantly to CTB binding to

Jurkat cells. In contrast, neither MAL-II, which recognizes α2-3-linked sialic acid, nor *Sambucus nigra* lectin (SNA), which recognizes α2-6-linked sialic acid, affected biotin-CTB binding to colonic epithelial cell lines (*Figure 7A–C*).

We tested whether inhibition of fucosylation affected production of CTB-glycoprotein crosslinked complexes. T84 or Colo205 cells were cultured with Ac$_4$ManNDAz and increasing concentrations of peracetylated 2-fluorofucose (2F-Fuc), a metabolic inhibitor of fucosylation (*Rillahan et al., 2012*). As the 2F-Fuc concentration increased, reduced amounts of the CTB-glycoprotein crosslinked complex were observed (*Figure 8A,B*), indicating that fucosylation is required for SiaDAz-dependent CTB crosslinking to glycoproteins. Next, we assessed whether fucosylation was required for the direct interaction between CTB and CEACAM5. T84 cells were cultured with Ac$_4$ManNDAz and 2F-Fuc, then crosslinking to CTB was performed. CEACAM5 was isolated by immunoprecipitation and probed with anti-CTB to assess CTB-CEACAM5 crosslinking (*Figure 8C*). Less crosslinked CTB-CEACAM5 was observed when fucosylation was inhibited. One possible explanation for this result is that the CEACAM5-CTB interaction is mediated by a fucosylated glycan. Additionally, the decreased crosslinking may reflect reduced overall binding of CTB to the cell surface when fucosylation is inhibited.

We tested whether inhibiting fucosylation affected overall binding of CTB to colonic epithelial cell lines. First, we confirmed the effectiveness of 2F-Fuc to reduce cell surface fucose in T84 cells by showing that it caused a reduction in binding of UEA-1 (*Figure 8—figure supplement 1*). Then, T84 cells were cultured with or without inhibitory concentrations of 2F-Fuc, followed by incubation with increasing concentrations of biotin-CTB. The amount of biotin-CTB bound to the cell surface was measured by an ELISA method (*Figure 8D*). Inhibition of fucosylation reduced the plateau value for biotin-CTB binding to T84 cells. We also used fluorescence microscopy to measure binding of Alexa Fluor 647-labeled CTB to T84 cells cultured with or without 2F-Fuc. Cells cultured with the fucosylation inhibitor displayed a reduction in Alexa Fluor 647-CTB binding (*Figure 8E* and *Figure 8—figure supplement 2*). Similarly, Colo205 cells were cultured with 2F-Fuc and CTB binding was measured by flow cytometry (*Figure 8F*). Inhibition of fucosylation reduced CTB binding to Colo205 cells, too. In contrast, Colo205 cells cultured with a sialylation inhibitor, 3F-NeuAc, (*Rillahan et al., 2012*) displayed enhanced binding to CTB, consistent with a small increase in binding of a fucose-recognizing lectin that also occurs with inhibition of sialylation (*Figure 8F* and *Figure 8—figure supplement 3*). We conclude that recognition of fucosylated structures is an important and general mechanism for CTB binding to colonic epithelial cells. The data regarding sialylation are less clear-cut and additional experiments will be required to determine whether sialic acids are components of the glycan motifs that CTB recognizes on colonic epithelial cells.

## Fucosylation and protein glycosylation mediate CTB internalization and host cell intoxication

Having discovered that fucosylation and protein glycosylation are important factors mediating CTB binding on the surface of colonic epithelial cell lines, we next tested whether fucosylated glycoproteins were involved in the mechanism by which CT intoxicates host cells. We used an in-cell ELISA method to measure the effects of inhibitors of glycosylation on the uptake of biotin-CTB by T84 cells. Cells were first cultured with glycosylation inhibitors, as described above. To quantify internalized biotin-CTB, cells were incubated in the presence of biotin-CTB at 37°C for the indicated times, then unbound biotin-CTB was washed away and the cells were rapidly cooled to 4°C to arrest internalization. Remaining surface-bound biotin-CTB was blocked with avidin, and also removed by acid washing. After permeabilizing cells, internalized biotin-CTB was measured by ELISA (*Figure 9A*). Total surface bound biotin-CTB was simultaneously assessed by maintaining control cells at 4°C (no endocytosis) (*Figure 9—figure supplement 1*) and yielded values consistent with other measurements (*Figures 4B,C*, *Figure 8D,E*). The amount of biotin-CTB internalized by cells treated with NB-DGJ was indistinguishable from that observed for control cells (*Figure 9A*), suggesting that gangliosides do not mediate the majority of biotin-CTB internalization. Kifunensine caused a slight enhancement in internalization, suggesting that *N*-linked glycosylation of proteins might serve as an impediment to CTB internalization. Culturing cells with either benzyl-α-GalNAc or 2F-Fuc each nearly abolished internalization, implying that fucosylated and/or *O*-linked glycoproteins may play functional roles in biotin-CTB internalization.

Next, we examined the effects of inhibitors of glycosylation on CT-induced elevation of cAMP, a later step in host cell intoxication. T84 cells were cultured in monolayers in the presence of glycosylation inhibitors. First, we evaluated whether inhibitors of glycosylation caused off-target effects on adenylate cyclase activity. Indeed, both benzyl-α-GalNAc and kifunensine caused reductions in the amount of cAMP that accumulated in response to stimulation with forskolin (*Figure 9—figure supplement 2A*), and with vasoactive intestinal peptide (VIP; *Figure 9—figure supplement 2B*), and similarly reduced cAMP accumulation in response to CT (*Figure 9—figure supplement 2C*). While the effects of benzyl-α-GalNAc and kifunensine on adenylate cyclase activity make it difficult to interpret how these inhibitors modulate CT intoxication, interpreting the effects of NB-DGJ and 2F-Fuc is more straightforward since neither affected cAMP accumulation in response to forskolin or VIP (*Figure 9—figure supplement 2A,B*).

CT holotoxin was incubated with cells at 37°C for 1 hr, then cAMP accumulation was measured. 2F-Fuc treatment resulted in dramatically decreased accumulation of cAMP, while treatment with NB-DGJ had a more moderate effect (*Figure 9B*). Thus, host cell fucosylation is important for intoxication by CT. Since 2F-Fuc treatment also causes reduced CTB binding and cell entry, the most likely explanation for this result is that reduced fucosylation leads to less CT entering cells, and thereby reduces host cell intoxication. In contrast, NB-DGJ does not dramatically affect CTB binding or internalization, so the explanation for this result is less clear. One possibility is that there is a small amount of GM1 in the cells, which is capable of mediating host cell intoxication via pathway that operates in parallel to the fucosylated glycoproteins. A second possibility is that GM1 is not required for the initial steps of CT binding and internalization, but is required for later steps in intoxication. A final possibility is that GM1 is not essential, but other glucosylceramide glycolipids, which are also reduced due to NB-DGJ treatment, play roles in the trafficking and intoxication process. Notably, CT intoxication appeared to be sensitive to brefeldin A in both untreated cells and in cells that were cultured with NB-DGJ (*Figure 9—figure supplement 3*), implying that in both cases, CT intoxication of host cells occurs via retrograde transport through the secretory pathway, consistent with other studies (*Lencer, 2003*).

## CTB recognizes fucosylation and glycoproteins in normal gut epithelia

The results implicating fucosylation and protein glycosylation in CTB binding and internalization are unexpected since the ganglioside GM1 is generally accepted to be the sole receptor for CT. The experiments reported here were performed in T84 and Colo205 cell lines, both colorectal cancer cell lines. While T84 cells are widely used as a model for host cell intoxication by CT, we wondered whether fucosylation and protein glycosylation might also function in CTB binding to normal colonic epithelial cells. To investigate, we used a human colonic epithelial cell line (HCEC) derived from normal human colon cells and immortalized by expression of Cdk4 and hTERT (*Roig et al., 2010*). HCEC cells were cultured with inhibitors of glycosylation (NB-DGJ, kifunensine, benzyl-α-GalNAc, or 2F-Fuc), then CTB binding was measured by an in-cell ELISA method (*Figure 10A*). CTB binding to HCECs was low overall and unaffected by culturing the cells with either kifunensine or NB-DGJ. Culturing HCECs with benzyl-α-GalNAc resulted in an increase in CTB binding, while culturing HCECs with 2F-Fuc resulted in a decrease in CTB binding. These results suggest that GM1 is not an important determinant for CTB binding to HCECs, and point to a contribution from fucosylated structures, potentially displayed on glycoproteins.

While colonic epithelial cells are commonly used as a model system for CT studies, the primary physiological site for CT action is the small intestine. We assessed whether CTB-binding glycoproteins were present in epithelial tissue from different parts of the normal gut. Lysates from T84 cells, Colo205 cells, human small intestine (duodenum, ileum, and jejunum), and human colon were separated by SDS-PAGE and probed with CTB-HRP (*Figure 10B*). Discrete CTB-reactive bands were found in all samples, with the apparent mobility of these species depending on the lysate source. Thus, CTB-binding glycoproteins are present in normal human epithelial tissue, both from the small intestine and the colon.

Finally, we assessed whether fucosylation plays a role in CTB binding to freshly isolated colonic epithelium. Epithelial cells were obtained from unaffected mucosa of colon adenocarcinoma patients undergoing curative tumor resection. Binding of CTB to colonic epithelial cells (EpCAM[+] CD45[-]) was significantly blocked by AAL (*Figure 10C*) and free fucose (*Figure 10D*), but not by glucose or

galactose, similar to the results observed with T84 and Colo205 cells. We conclude that fucosylated structures make a significant contribution to CTB binding to normal colonic epithelial tissue.

## Discussion

In this study we show that glycoproteins mediate a large fraction of CTB binding to colonic epithelial cell lines. In contrast, in the Jurkat T cell line, GM1 is the main CTB binding ligand. Further, the results presented here implicate glycoproteins as functional receptors that can mediate toxin internalization. Our findings are in conflict with dogma, (*Foster and Baron, 1996*) but not inconsistent with previous results. The earliest studies identifying GM1 as the CT receptor were based on inhibition assays and gain-of-function experiments. Indeed, CTB binds GM1 with high affinity, (*Kuziemko et al., 1996*) so it is not surprising that GM1 is an effective inhibitor of CTB binding and that adding GM1 to cells leads to enhanced CTB binding. However, experimentally addressing how loss of GM1 affects CTB binding and host cell intoxication has been more difficult. Use of selective inhibitors of glycosphingolipid biosynthesis allowed two groups to observe that a portion of CTB binding is GM1-independent (*Platt et al., 1997*; *Blank et al., 2007*). Our data extend this observation, showing that CTB binding is mostly GM1-independent in cell types more closely related to the physiological site of CT action. Further, we report evidence that CTB binds directly to glycoproteins, relying on glycan recognition motifs. Our observation that inhibition of ganglioside biosynthesis does not affect CTB binding to colonic epithelial cell lines is consistent with the low level of GM1 in these cells. Similarly, the normal human intestinal epithelia contains very little GM1, (*Breimer et al., 2012*) suggesting that glycoproteins could be important contributors to CTB binding during *Vibrio cholerae* infection.

In addition to its critical role in human disease, host cell intoxication by CT is an important model system for studying endocytic mechanisms (*Chinnapen et al., 2007*). CT has been shown to enter cells through both clathrin- and caveolin-dependent mechanisms, as well a clathrin- and caveolin-independent mechanism (*Orlandi, 1998*; *Torgersen et al., 2001*; *Massol, 2004*; *Howes et al., 2010*). Studies to identify the dominant endocytic mechanism have yielded variable results, raising the possibility that CT uses different endocytic mechanisms to enter different cell lines. The existence of multiple glycoprotein receptors provides a plausible explanation for the use of multiple endocytic pathways and for cell type differences. The exact structure of the glycan might also play a functional role in the endocytic mechanism, as modification of proteins with a GM1 ganglioside is sufficient to engender CT binding, but not host cell intoxication (*Pacuszka and Fishman, 1990*). In addition, a long-standing conundrum about CT internalization is how binding to GM1 located in the outer leaflet of the plasma membrane can trigger endocytosis. Our results show that CTB can bind glycoproteins, offering a possible mechanism for receptor-mediated endocytosis.

While we show that CEACAM5 binds directly to CTB through a glycan-dependent interaction, it seems unlikely that CEACAM5 is an important physiological receptor for CT. Our detection of CEACAM5 in T84 cells likely reflects the increased CEACAM5 expression and altered CEACAM5 glycosylation often found in colorectal cancer (*Saeland et al., 2012*). In addition, while *N*-linked glycosylation is critical for the CTB-CEACAM5 interaction, the majority of CTB binding to T84 cells depends on *O*-linked glycosylation of proteins. Another protein previously reported to interact with CTB in pig enterocytes is sucrase-isomaltase (SI) (*Hansen et al., 2005*); however, we did not detect any peptides corresponding to SI in our proteomics analysis of the CTB crosslinked complex. Intriguingly, though, SI is subject to GalNAc-type *O*-linked glycosylation, which controls its association with detergent-insoluble membrane microdomains (*Alfalah et al., 1999*). We speculate that certain GalNAc-type *O*-linked glycans may serve as determinants for both CTB recognition and membrane microdomain targeting, potentially explaining why CTB binding fractionates with detergent-insoluble material, an observation that is typically interpreted to result from the binding of CTB to GM1.

The demonstration that glycoproteins can be important contributors to CTB binding has implications for the use of CTB to study the organization of lipids in the plasma membrane. Indeed, differences in observed diffusion rates for bound CTB may reflect the identity of the CTB ligand, rather than the fluidity of the membrane in which it resides. Day and Kenworthy noticed that CTB bound to COS-7 cells diffused surprisingly slowly for a lipid-bound protein and speculated that an interaction with a protein might slow its diffusion (*Day et al., 2012*). Our results predict that CTB diffusion rates

will differ in different cell types, with rapid diffusion in GM1-rich cell lines, like Jurkat cells, and slower diffusion in GM1-deficient cell lines, such as T84 and Colo205 cells.

The functional assays reported here do not directly address whether glycoproteins serve as functional CT receptors in the normal human gut. However, a consistent theme among the results reported here is that fucose plays an important role in CTB binding, both to cell lines and ex vivo to freshly isolated human epithelial cells. While the lectin blocking studies demonstrate overlap among the sets of glycans recognized by AAL and CTB, additional work is needed to determine the exact structure of the fucosylated glycans to which CTB prefers to bind. More broadly, identification of fucose as an important determinant of CTB binding adds to a growing body of literature that implicates fucose in host-microbe discourse in the gut (*Pacheco et al., 2012*; *Pickard et al., 2014*).

In summary, we demonstrate that fucosylated glycoproteins mediate a large portion of CTB binding to human colonic epithelial cell lines, that fucosylated glycoproteins play an important role in the mechanism by which CTB enters T84 cells, and that entry of CT into T84 cells via a fucose-dependent mechanism is on-pathway to host cell intoxication. These findings raise the possibility that fucose-containing or -mimicking molecules may have utility in cholera therapy. In addition, the observation that CTB binds to cell surface molecules other than GM1 implies that caution should be applied in the interpretation of experiments where CTB is used to visualize membrane microdomain structures.

# Materials and methods

## General
### Chemicals
Ac$_4$ManNDAz was synthesized in very good purity according to TLC and $^1$H-NMR as described previously (*Tanaka and Kohler, 2008*; *Bond et al., 2009*). Dimethyl sulfoxide (DMSO) was purchased from Sigma (St. Louis, MO) (catalog no. D2650). N-(n-Butyl)deoxygalactonojirimycin (NB-DGJ; 98% pure) was purchased from Santa Cruz Biotechnology (Dallas, TX) (catalog no. sc-221974); stock concentrations were made at 5 mg/mL in water then sterile filtered. 1-Deoxymannojirimycin hydrochloride ($\geq$ 98% pure) was purchased from Sigma (catalog no. D9160); stock concentrations were made at 400 mM in water then sterile filtered. Kifunensine ($\geq$ 98% pure) was purchased from Sigma (catalog no. K1140); stock concentrations were made at 1 mg/mL in water then sterile filtered. Benzyl-2-acetamido-2-deoxy-α-d-galactopyranoside (benzyl-α-GalNAc; $\geq$ 97% pure) was purchased from Sigma (catalog no. B4894); stock concentrations were made at 1 M in DMSO. 2-Fluoro-peracetyl-fucose (2F-Fuc; 98.8% pure) was purchased from EMD Millipore (Darmstadt, Germany) (catalog no. 344827); stock concentrations were made at 200 mM in DMSO. 3-Fluoro-peracetyl-NeuAc (3F-NeuAc; $\geq$ 98% pure) was purchased from EMD Millipore (catalog no. 566224); stock concentrations were made at 200 mM in DMSO. l-(−)-Fucose ($\geq$ 99% pure) was purchased from Sigma (catalog no. F2252). d-(+)-Glucose ($\geq$ 99.5% pure) was purchased from Sigma (catalog no. G7021). N-Acetyl-D-glucosamine ($\geq$ 95% pure) was purchased from Sigma (catalog no. A3286). d-(+)-Galactose ($\geq$ 99% pure) was purchased from Sigma (catalog no. G0750). N-Acetyl-D-galactosamine (~98% pure) was purchased from Sigma (catalog no. A2795). N-Acetylneuraminic acid (min 98% pure) was purchased from Carbosynth (Berkshire, UK) (catalog no. MA00746). Bovine serum albumin (BSA, Fraction V, heat shock treated) was purchased from both Fisher (Waltham, MA) (catalog no. BP1600) and Sigma (catalog no. A9647). Skim milk powder was purchased from EMD Millipore (catalog no. 115363). Carbohydrate-free blocking solution was purchased from Vector Laboratories (Burlingame, CA) (catalog no. SP-5040). Paraformaldehyde (formaldehyde) aqueous solution (20%) was purchased from Electron Microscopy Sciences (Hatfield, PA) (catalog no. 15713). Forskolin ($\geq$ 98% pure) was purchased from Sigma (catalog no. F6886); stock concentrations were made at 10 mM in DMSO. Vasoactive Intestinal Peptide (VIP) ($\geq$ 95% pure) was purchased from Anaspec (Fremont, CA) (catalog no. AS-22873); stock concentrations were made at 2 mM in water then sterile filtered.

### Cell culture
The following reagents for general cell culture use were purchased from Life Technologies/Gibco (Carlsbad, CA): penicillin-streptomycin (catalog no. 15140), fetal bovine serum (FBS) (catalog no. 16000), heat inactivated fetal bovine serum (HI-FBS) (catalog no. 10082), TrypLE express enzyme

with phenol red (catalog no. 12605), and distilled water (catalog no. 15230). Dulbecco's Phosphate Buffered Saline (DPBS) was purchased from Sigma (catalog no. D8537). Jurkat cells (obtained from Kim Orth, UT Southwestern Medical Center) were maintained in RPMI 1640 medium supplemented with 2 mM L-glutamine (Gibco, catalog no. 11875) and 10% (v/v) HI-FBS. T84 cells (ATCC, Manassas, VA) were maintained in DMEM/F-12 medium supplemented with 2.5 mM L-glutamine, 15 mM HEPES (Gibco, catalog no. 11330), 5% (v/v) FBS, and penicillin-streptomycin. T84 cells were not used past passage number 45. Colo205 wild-type and SimpleCells (*Steentoft et al., 2011*) were maintained in RPMI 1640 medium supplemented with 2 mM L-glutamine, 10% (v/v) FBS, and penicillin-streptomycin. hCMEC/D3 cells (*Weksler, 2005*) (obtained from Babette Weksler, Weill Cornell Medical College) were cultured in endothelial basal medium-2 (EBM-2) (Lonza, Walkersville, MD) (catalog no. 00190860) supplemented with 5% (v/v) FBS, 1.4 µM hydrocortisone (Sigma, catalog no. H0135), 5 µg/mL ascorbic acid (Sigma, catalog no. A4544), 1:100 dilution of chemically defined lipid concentrate (Life Technologies, catalog no. 11905–031), 10 mM HEPES, and 1 ng/mL human basic fibroblast growth factor (Sigma, catalog no. F0291). Human bronchial epithelial cells (HBEC) (*Ramirez, 2004*) (obtained from Jerry Shay, UT Southwestern Medical Center) were cultured in keratinocyte serum-free medium (Gibco, catalog no. 17005–042) containing bovine pituitary extract (BPE) and recombinant human epidermal growth factor (EGF 1–53) in tissue culture plates coated with porcine gelatin (Sigma, catalog no. G1890). Human colonic epithelial cells (HCEC) (*Roig et al., 2010*) (obtained from Jerry Shay, UT Southwestern Medical Center) were cultured in Primaria flasks (BD Biosciences, San Jose, CA) in basal X medium (HyClone, GE Healthcare) (Logan, UT) supplemented with EGF (25 ng/mL; PeproTech, Inc, Rocky Hill, NJ), hydrocortisone (1 µg/mL; Sigma), insulin (10 µg/mL, Sigma), transferrin (2 µg/mL; Sigma), sodium selenite (5 nM; Sigma), 2% (v/v) cosmic calf serum (HyClone, GE Healthcare), and gentamicin sulfate (50 µg/mL) (Gemini Bio-Products, West Sacramento, CA) (*Roig et al., 2010*). All cell lines (excluding HCEC cells) were maintained at 37°C, 5% carbon dioxide in a water-saturated environment; the HCEC cells were maintained in 2–5% oxygen and 7% carbon dioxide. The Countess automated cell counter (Life Technologies) was used for cell counting.

## Sequencing C1GALT1C1

To sequence the C1GALT1C1 gene, genomic DNA was isolated from 4 million Jurkat and T84 cells using the PureLink genomic DNA mini kit (Life Technologies, catalog no. K1820) according to the manufacturer's instructions. The C1GALT1C1 gene is composed of a single exon and was thereby amplified by the polymerase chain reaction using the forward primer 5' ATGCTTTCTGAAAGCAGCTCC 3' and the reverse primer 5' TCAGTCATTGTCAGAACCATTTGG 3'. The PCR products were purified using the PureLink PCR purification kit (Life Technologies, catalog no. K3100) according to the manufacturer's instructions and submitted to the UT Southwestern Sanger Sequencing Core.

## Cholera toxin

Cholera toxin B subunit used for photocrosslinking experiments was purchased from Sigma (catalog no. C9903). Biotin-conjugated cholera toxin subunit B used for mass spectrometry and binding experiments was purchased from Life Technologies (catalog no. C-34779). Alexa Fluor 647-conjugated cholera toxin subunit B used for fluorescence microscopy experiments was purchased from Life Technologies (catalog no. C-34778). For flow cytometry experiments with patient samples, CTB was conjugated with iFlur647 according to manufacturer's instructions (ReadiLink; Bio-Rad). Cholera toxin (azide-free) from *Vibrio cholerae* used for cAMP experiments was purchased from List Biological Laboratories (Campbell, CA) (catalog no. 100B). Horseradish peroxidase (HRP)-conjugated cholera toxin subunit B used for immunoblot experiments was purchased from Life Technologies (catalog no. C-34780).

## Lectins and antibodies

The sources of the antibodies used for immunoblotting are as follows: anti-beta subunit cholera toxin antibody (Abcam, Cambridge, MA) (catalog no. ab34992), anti-cholera toxin antibody (Sigma, catalog no. C3062), anti-CEACAM5 antibody (clone Col-1) (Life Technologies, catalog no. 18–0057), anti-CD44 antibody (156-3C11) (Cell Signaling Technology, Boston, MA) (catalog no. 3570), anti-LAMP1 antibody (clone 25/Lamp-1) (BD Biosciences, catalog no. 611042), anti-α-tubulin antibody

(Sigma, catalog no. T6199), and anti-β-actin antibody (8H10D10) (Cell Signaling Technology, catalog no. 3700). Goat anti-Rabbit IgG-HRP conjugate (catalog no. 65–6120) and goat anti-Mouse IgG-HRP conjugate (catalog no. 62–6520) secondary antibodies were purchased from Life Technologies.

Most lectins were purchased from Vector Laboratories. Biotinylated lectins used for flow cytometry consisted of the following: biotinylated concanavalin A (Con A) (catalog no. B-1005), biotinylated peanut agglutinin (PNA) (catalog no. B-1075), biotinylated *Ulex europaeus* agglutinin I (UEA I) (catalog no. B-1065), biotinylated *Lotus tetragonolobus* lectin (LTL) (catalog no. B-1325), biotinylated *Aleuria aurantia* lectin (AAL) (catalog no. B-1395), biotinylated *Maackia amurensis* lectin II (MAL II) (catalog no. B-1265), and biotinylated *Sambucus nigra* lectin (SNA, EBL) (catalog no. B-1305). Unconjugated lectins used for blocking experiments were reconstituted in the buffer recommended by the manufacturer, and consisted of the following: unconjugated *Ulex europaeus* agglutinin I (UEA I) (catalog no. L-1060), unconjugated *Lotus tetragonolobus* lectin (LTL) (catalog no. L-1320), unconjugated *Aleuria aurantia* lectin (AAL) (catalog no. L-1390), unconjugated *Lens culinaris* agglutinin (LCA) (catalog no. L-1040), unconjugated *Maackia amurensis* lectin II (MAL II) (catalog no. L-1260), and unconjugated *Sambucus nigra* lectin (SNA, EBL) (catalog no. L-1300). Unconjugated *Erythrina cristagalli* lectin (ECL) was purchased from Sigma (catalog no. L5390).

## SiaDAz-mediated CTB crosslinking

For photocrosslinking of CTB to Jurkat T cells (a suspension cell line), 2 million cells were first seeded in 10 mL media into 10-cm tissue culture plates treated with either vehicle (evaporated ethanol) or 100 μM Ac$_4$ManNDAz. After culturing for 72 hr, cells were counted, re-suspended in fresh media to a concentration of 5 million cells/mL, transferred to two separate multiwell tissue culture plates (for –/+ UV), and CTB (Sigma) was added at a concentration of 2.5 μg/mL. The toxin was allowed to bind to the cell surface for 45 min at 4°C in the dark. The cells were then either kept at 4°C for an additional 45 min (for the – UV samples) or irradiated on an ice/water bath for 45 min (for + UV samples) at 365 nm (UVP, XX-20BLB lamp). The cells were then collected, washed with Dulbecco's Phosphate Buffered Saline (DPBS), and lysed on ice for 30–60 min in RIPA buffer (50 mM TrisHCl, pH 8.0, 150 mM NaCl, 1% (v/v) NP-40, 0.5% (v/v) sodium deoxycholate, 0.1% (w/v) sodium dodecyl sulfate (SDS) and a protease inhibitor cocktail (Roche, Indianapolis, IN) (catalog no. 11836170001)). The lysate was pelleted at 20,817$g$ for 10 min at 4°C to clear the insoluble debris, and the supernatant was retained for further immunoblot analysis. Protein content was quantified with a BCA assay kit (Thermo Scientific Pierce Protein Biology, Waltham, MA) against a BSA standard curve for normalization. For resolution on a high percentage (15%) Tris-glycine gel, 9 μg of lysate was denatured in 2X SDS loading dye (100 mM TrisHCl pH 6.8, 4% (w/v) SDS, 0.04% (w/v) bromophenol blue, 20% (v/v) glycerol, and 10% (v/v) 2-mercaptoethanol) for 5 min at 90°C. For resolution on a lower percentage (6%) Tris-glycine gel, 12 μg of lysate was denatured in 4X SDS loading dye (200 mM TrisHCl pH 6.8, 8% (v/v) SDS, 0.08% (w/v) bromophenol blue, 40% (v/v) glycerol, and 40 mM DTT) for 5 min at 90°C. The samples were separated by SDS-PAGE and transferred to a PVDF membrane (EMD Millipore, catalog no. IPVH00010). The blots were probed overnight at 4°C for anti-CTB (Abcam, 1:10,000 dilution) in 5% (w/v) non-fat milk in TBST. The blots were then probed with a goat anti-rabbit HRP conjugated secondary antibody (1:5000 dilution) for 1 hr at room temperature, and developed using the SuperSignal West Pico Chemiluminescent Substrate (Thermo Scientific, catalog no. 34080) and X-ray film. Membranes were stripped in mild stripping buffer (200 mM glycine, 0.1% (w/v) SDS, 1% (v/v) Tween-20, pH 2.2) for 45 min at 37°C before re-probing for the loading control anti-α-tubulin (Sigma, 1:10,000 dilution) for 1 hr at room temperature. The blots were then probed with a goat anti-mouse HRP conjugated secondary antibody (1:5000 dilution) for 1 hr at room temperature, and developed using the SuperSignal West Pico Chemiluminescent Substrate and X-ray film.

For photocrosslinking of CTB to human colonic epithelial cell lines (both adherent cell lines), either 250,000 T84 cells or 500,000 Colo205 cells were seeded in 2 mL of media into two separate 6-well tissue culture plates (for –/+ UV) that were treated with either vehicle (evaporated ethanol) or Ac$_4$ManNDAz to a final concentration of 100 μM. When used, glycosylation inhibitors were also added at the time of seeding to achieve these final concentrations: 50 μM NB-DGJ, 2 mM deoxymannojirimycin, 1 μg/mL kifunensine, 2 mM benzyl-α-GalNAc, and 5 – 200 μM 2F-Fuc. Experimental samples were compared to the appropriate vehicle-only control: water for NB-DGJ, deoxymannojirimycin, and kifunensine; DMSO for benzyl-α-GalNAc and 2F-Fuc. After culturing for 72 hr, the media

in each well was replaced with 1 mL fresh media containing approximately 4.5 μg of CTB (Sigma). The toxin was allowed to bind to the cell surface for 45 min at 4°C in the dark. The cells were then either kept at 4°C for an additional 45 min (for – UV samples) or irradiated on an ice/water bath for 45 min (for + UV samples) at 365 nm. Cells were then washed with DPBS, collected into RIPA lysis buffer with a cell scraper, and incubated on ice for 30 – 60 min. The lysate was centrifuged at 21,000$g$ for 10 min at 4°C to remove insoluble debris, and the supernatant was retained for separation on both a higher (15%) and lower (ranging from 6 – 7.5%) percentage gel, as described previously for Jurkat T cells. The samples were then transferred to a PVDF membrane, and the blots were probed overnight at 4°C for either anti-CTB (Abcam, 1:10,000 dilution) or anti-CT (Sigma, 1:10,000 dilution) as indicated in the figure labels. Membranes were stripped and re-probed for the loading control anti-α-tubulin as described previously. To ensure the glycosyltransferase inhibitors were effective in T84 cells, samples were probed for 1 hr at room temperature for either anti-LAMP-1 (BD Biosciences, 1:5000 dilution) or anti-CD44 (Cell Signaling Technology, 1:2500 dilution). Photocrosslinking of CTB to hCMEC/D3 or HBEC cell lines was carried out in an analogous manner, with the exception that 200,000 cells were seeded into 6-well tissue culture plates, and approximately 1.1 μg of CTB (Sigma) was added to each well.

## Ganglioside analysis

### Isolation of glycosphingolipids

T84, Colo205 and Jurkat cells were cultured with either vehicle or 50 μM NB-DGJ for 3 d. For each sample, 60 million cells were harvested and washed with DPBS. Cell pellets were stored at –80°C until lyophilization. The lyophilized materials were extracted in a Soxhlet apparatus with chloroform and methanol (2:1, by volume) for 24 hr, followed by chloroform and methanol (1:9, by volume) for 24 hr. The two lipid extracts from each cell preparation were pooled, subjected to mild alkaline hydrolysis. Then the extracts were separated on a 1 g silicic acid column, eluted first with 5% (by volume) methanol in chloroform to remove non-polar material, and thereafter eluted with 75% (by volume) methanol in chloroform and 100% methanol. The two latter fractions were combined and dried.

### Radiolabeling

Aliquots of 100 μg of CTB and *Erythrina cristagalli* lectin (Sigma-Aldrich), were labeled with [125]I by the Iodogen method according to the manufacturer's instructions (Pierce, Rockford, IL), giving approximately 5000 cpm/μg protein.

### Reference glycosphingolipids

Reference glycosphingolipids were isolated and characterized by mass spectrometry and proton NMR as described (*Karlsson, 1987*).

### Thin-layer chromatography

Partially purified glycosphingolipids were redissolved in 500 μL chloroform/methanol (2:1, by volume), and 4–16 μL of each fraction were applied on aluminum-backed silica gel 60 HPTLC plates (Merck, Kenilworth, NJ), along with reference GM1 ganglioside (1–5 ng/lane), and separated using chloroform/methanol/water (60:35:8, by volume) as solvent. After drying, this chromatogram was probed with [125]I-labeled CTB, as described below. Thereafter, the fractions were dried and redissolved in 200 μL chloroform/methanol, and thin-layer chromatograms were prepared once again using 4–16 μL of each fraction and reference GM1 ganglioside (1–5 ng/lane). After drying, these chromatograms were probed with [125]I-labeled CTB and [125]I-labeled *E. cristagalli* lectin, as described below. Finally, the fractions were dried, redissolved in 80 μL chloroform/methanol, and 8 μl of each fraction and reference gangliosides (2 μg/lane) were separated on thin-layer plates as above. After drying, this chromatogram was stained with resorcinol (*Svennerholm and Fredman, 1980*).

Thin-layer chromatogram binding assays were performed as described (*Teneberg et al., 1994*; *Angström et al., 1994*). Dried chromatograms were dipped in diethylether/*n*-hexane (1:5 v/v) containing 0.5% (w/v) polyisobutylmethacrylate for 1 min, dried, and then blocked with BSA/PBS (PBS containing 2% (w/v) bovine serum albumin and 0.1% (w/v) NaN$_3$) for 2 hr at room temperature. Thereafter, the plates were incubated with [125]I-labeled CTB (1–8 × 10$^6$ cpm/ml), or *E. cristagalli*

lectin ($2 \times 10^6$ cpm/ml), diluted in BSA/PBS for another 2 hr at room temperature. After washing six times with PBS, and drying, the thin-layer plates were autoradiographed for 2–12 hr using XAR-5 x-ray films (Eastman Kodak, Rochester, NY).

## CTB binding assays

### Flow cytometry

For experiments involving inhibitor-treated cells, 2 mL of cells were seeded into a 6-well tissue culture plate at a density of 200,000 cells/mL for Jurkat and 125,000 cells/mL for Colo205. Inhibitors were then added to achieve the following final concentrations: 50 µM NB-DGJ, 1 µg/mL kifunensine, 2 mM benzyl-α-GalNAc, 200 µM 2F-Fuc, and 200 µM 3F-NeuAc. Experimental samples were compared to the appropriate vehicle-only control: water for NB-DGJ, deoxymannojirimycin, and kifunensine; DMSO for benzyl-α-GalNAc, 2F-Fuc, and 3F-NeuAc. After culturing for 72 hr, cells were harvested by centrifugation and transferred in DPBS to a v-bottom 96-well plate (Costar, catalog no. 3897) as follows. Jurkat cells were pelleted by centrifugation, resuspended in DPBS, and ~450,000 cells in 200 µL were transferred per well. Colo205 cells were removed from tissue culture plates by addition of 0.7 mL trypsin for 2 min at 37°C, pelleted by centrifugation, resuspended in DPBS, and ~300,000 cells in 200 µL were transferred per well. One in the v-bottom plate, cells were washed twice by resuspension in 200 µL cold DPBS containing 0.1% (w/v) BSA (DPBS/BSA) followed by centrifugation at 730g for 4 min at 4°C. Cells were then incubated 30 min on ice with diluted biotinylated versions of CTB or lectins at the following concentrations: 50 µL of a 4 µg/mL dilution was added per well to Jurkat cells and 50 µL of a 10 µg/mL dilution was added per well to Colo205 cells. The cells were washed twice in 200 µL with cold DPBS/BSA, then incubated for 30 min on ice with 50 µL of a 7.7 µg/mL dilution of fluorescein (DTAF) streptavidin (Jackson ImmunoResearch, West Grove, PA) (catalog no. 016-010-084) added per well. The cells were again washed twice, then analyzed on a FACSCalibur flow cytometer (BD Biosciences, UT Southwestern Flow Cytometry Core Facility). The live population of cells was gated based on forward and side scatter emission, and degree of binding was determined by fluorescence intensity on the FL1 emission channel. Background biotin signal was detected with cells treated with DPBS/BSA and DTAF-streptavidin only (indicated as BSA control in plots of flow cytometry data).

For experiments in which free monosaccharides were used as blocking agents, Jurkat and Colo205 cells were harvested from 10 cm tissue culture plates seeded at either 2 million cells/10 mL and cultured for 2 d or 1 million cells/10 mL and cultured for 3 d. Cells were harvested and transferred to a v-bottom 96-well plate as follows. Jurkat cells were pelleted by centrifugation, resuspended in DPBS, and ~450,000 cells in 200 µL were transferred per well. Colo205 cells were removed from the tissue culture plate by adding 1.5 mL of trypsin for 2 min at 37°C, pelleted by centrifugation, resuspended in DPBS, and ~350,000 cells in 200 µL were transferred per well. Cells were washed twice with 200 µL cold DPBS/BSA, then incubated 30 min on ice with biotinylated versions of CTB or lectins diluted in DPBS/BSA containing the indicated concentrations of free fucose, glucose, GlcNAc, galactose, GalNAc, or Neu5Ac. The cells were washed twice in 200 µL cold DPBS/BSA then incubated for 30 min on ice with DTAF-streptavidin. The cells were again washed twice, then analyzed on a FACSCalibur flow cytometer. The background median fluorescence intensity (MFI) for the BSA control sample was subtracted from each sample, and the MFI for the sample with no monosaccharide was normalized to 100% bound. Averages of 3 independent experiments and their standard deviations are depicted in bar graphs.

For lectin blocking experiments, Jurkat and Colo205 cells were harvested from 10-cm tissue culture plates and transferred to a v-bottom 96-well plate as described above. Next, a 30 min "pre-toxin blocking step" was introduced, in which 50 µL of un-conjugated lectin diluted in DPBS/BSA to the indicated concentrations (ranging from 10–100 µg/mL) was added per well. After washing the cells twice with 200 µL cold DPBS/BSA, binding of biotinylated CTB, detection with DTAF-streptavidin, and analysis on a FACSCalibur flow cytometer occurred as described previously.

The effect of 2F-Fuc and 3F-NeuAc treatment on T84 cell surface fucosylation and sialylation, respectively, was determined by flow cytometry. T84 cells were seeded in 2 mL media at a density of 50,000 cells/mL into a 6-well tissue culture plate, and the inhibitors were added to a final concentration of 100 µM. After culturing for 72 hr, T84 cells were removed from the tissue culture plate by adding 1 mL of trypsin for 5 min at 37°C, pelleted by centrifugation, resuspended in DPBS, and

~300,000 cells in 200 µL were transferred per well to a v-bottom 96-well plate. Cells were washed and probed with 50 µL of either a 20 µg/mL dilution of biotinylated UEA-I per well, a 10 µg/mL dilution of biotinylated MAL-II per well, or a 20 µg/mL dilution of biotinylated SNA per well. Lectin binding was detected with DTAF-streptavidin on a FACSCalibur flow cytometer.

## In-cell ELISA

For experiments with glycosylation inhibitors, T84 cells (25,000/well) were cultured in media in the absence or presence of each inhibitors (final inhibitor concentrations: 50 µM NB-DGJ, 1 µg/ml kifunensine, 2 mM benzyl-α-GalNAc, 100 µM 2F-Fuc; stock solutions of NB-DGJ and kifunensine were in water, benzyl-α-GalNAc and 2F-Fuc were in DMSO) in individual wells of a collagen-coated 96-well plate (Costar, catalog no. 9102) for 3 d. Cells were washed 2 times in cold PBS, and further incubated with various concentrations of biotinylated CTB in PBS4+ (1 mM $CaCl_2$, 1 mM $MgCl_2$, 0.2% (w/v) BSA, and 5 mM glucose) for 20 min on ice. Unbound biotin-CTB was washed away 3 times in cold PBS. Then cells were fixed with 4% paraformaldehyde for 10 min on ice and 20 min at room temperature. After 3 washes with PBS, cells were blocked with 1% BSA/PBS. Streptavidin-HRP conjugate (1:10000; Roche) was incubated for 1 hr to detect biotin-CTB. HRP activity was measured by a stopped colorimetric assay using ortho-phenylenediamine (OPD) as a substrate. Light absorption at 490 nm was determined with a Synergy Neo microplate reader (BioTek, Winooski, VT) and all values were corrected by light absorbance at 650 nm and normalized by total cell protein content (bicinchoninic acid assay, Pierce BCA protein assay kit, Pierce). For HCEC cells, 3000 cells were cultured in the same manner as T84 cells. Cells were washed 2 times in cold PBS, incubated with various concentrations of biotin-CTB for 30 min on ice. The detection procedure was the same as that used for T84 cells.

For experiments in which free monosaccharides were used as blocking agents, 25,000 T84 cells were added to individual wells of a collagen-coated 96-well plate (Costar) and were treated with variable concentrations of biotinylated CTB in PBS4+ containing 200 mM of fucose, glucose, galactose, GalNAc, or Neu5Ac. For free sugar titration, T84 cells were incubated with variable concentrations of the above listed free sugars in PBS4+ containing 10 µg/ml of biotinylated CTB. For lectin blocking experiments, $2.5 \times 10^4$ T84 cells were added to individual wells of a collagen-coated 96-well plate (Costar) and were treated with 40 µg/mL of the indicated lectin in PBS4+, then variable concentrations of CTB in PBS4+. Alternatively, variable concentrations of lectin were used, followed by 10 µg/ml of CTB.

## Immunofluorescence

For inhibitor experiments, T84 cells were plated on glass coverslips and cultured in media with DMSO (used as a control for both inhibitor and free sugar experiments), 4 µg/mL NB-DGJ, 1 µg/ml kifunensine, 2 mM benzyl-α-GalNAc, or 100 µM 2F-Fuc at 37˚C. After 72 hr, cells were washed twice in PBS and treated with 20 nM CTB labeled with Alexa Fluor 647 in PBS for 5 min at room temperature. For free sugar experiments, T84 cells plated on glass coverslips were treated with 100 mM fucose or glucose and 20 nM Alexa-Fluor 647-labeled CTB in PBS for 5 min at room temperature. For all experiments, after incubation with CTB, cells were washed twice in PBS and fixed in 4% paraformaldehyde for 15 min at room temperature. After fixation, cells were washed twice in PBS and mounted onto glass slides using Vectashield Mounting Medium with DAPI (Vector Laboratories, catalog no. H-1200).

All coverslips were imaged on a Zeiss LSM 510 META confocal microscope using a 63X objective. To image CTB-Alexa Fluor 647, a 633 nm laser line was used; for DAPI, a tunable NIR laser for multiphoton excitation was used at 790 nm. Image files were imported into ImageJ (NIH) for analysis. Cell boundaries were determined from bright field images, and resulting regions were used on background-subtracted fluorescent images to determine average fluorescence in each cell cluster. Experiments were performed in triplicate, with 10 fields of view imaged per treatment each time. Student t-test was used to determine difference in means.

## CTB blot

For CTB-HRP blotting, 15 µg of normal human small intestine lysates purchased from Protein Biotechnologies (Ramona, CA) (duodenum, catalog no. HN-27-1; jejunum, catalog no. HN-27-3; ileum,

catalog no. HN-27-2), 25 µg of normal human colon lysate (HN-06), 10 µg of T84 and Colo205 cell lysates was separated by 5% SDS-PAGE. After semi-dry transfer to PVDF membrane, the membrane was blocked with 1X Carbo-free reagent (Vector lab) including 0.05% (v/v) Tween-20 at room temperature for 1 hr, followed incubation with 1:5000 diluted HRP-conjugated CTB at 4°C overnight. After 3 washes with PBS, the membrane was incubated with SuperSignal West Femto Maximum Sensitivity Substrate for 1 min, then imaged with a ChemiDoc MP Imaging system (Bio-Rad, Hercules, CA).

## Proteomics analysis of CTB crosslinked complex

### Crosslinking

For crosslinking for proteomics experiments, 2 million T84 cells were seeded into ten 10-cm tissue culture plates with media containing 100 µM Ac$_4$ManNDAz. After culturing for 72 hr, media on each plate was replaced with 3 mL of fresh media; for those five plates that were to be irradiated, the media also contained 10 µg/mL biotin-CTB (Invitrogen) (*i.e.*, 30 µg per plate). Cells were incubated for 45 min at 4°C in the dark to allow binding to occur, then the "minus biotin-CTB" plates were kept at 4°C for an additional 45 min, and the "plus biotin-CTB" plates were irradiated on an ice/water bath for 45 min at 365 nm (UVP, XX-20BLB lamp). Cells were washed twice with 2 mL DPBS and each plate was collected into 500 µL hypotonic lysis buffer (10 mM TrisHCl pH 7.3, 10 mM MgCl$_2$, 1 mM EDTA, 1 mM EGTA, and a protease inhibitor cocktail) with a cell scraper. Lysates were incubated on ice for 15 min, then homogenized by extrusion through a 25 gauge needle for 4 min. The samples were centrifuged at 1000*g* for 12 min at 4°C. Supernatant was collected and centrifuged a second time, and the resulting post nuclear supernatant (PNS) was transferred to a 1.5 mL high-speed microcentrifuge tube (Beckman Coulter, Brea, CA) (catalog no. 357448). The samples were centrifuged at 100,000*g* for 1 hr at 4°C in a TLA-100.3 rotor to separate the cytosolic fraction (supernatant) from the membrane fraction (pellet). Proteomics analysis was performed by both in-gel digest and in-solution digest.

### In-gel digestion

The membrane fractions of each set of samples (*i.e.* – and + UV) were combined into a 500 µL total volume of NP-40 buffer (50 mM TrisHCl pH 8.0, 150 mM NaCl, 1.0% NP-40, and a protease inhibitor cocktail). Solubilization was initiated by gentle mixing with a pipet tip followed by incubation on ice for 1 hr. Protein content was quantified with a BCA assay kit (Pierce) using a BSA standard curve. 700 µg of membrane fraction was diluted to 0.5 µg/µL in NP-40 buffer, and incubated with 50 µL of streptavidin-agarose (Pierce, catalog no. 20349) overnight at 4°C with end-over-end rotation. The beads were washed 5 times with 1 mL NP-40 buffer containing 150 mM NaCl (low salt), followed by 5 washes with NP-40 buffer containing 500 mM NaCl (high salt). The beads were eluted for 7 min at 90°C with 35 µL of an SDS loading dye containing free biotin: 50 mM TrisHCl pH 6.8, 2.5% SDS (w/v), 10% glycerol (v/v), 0.1% bromophenol blue (w/v), 20 mM biotin, and 150 mM DTT. The supernatant was collected and separated on a 7.5% Mini-PROTEAN TGX gel (Bio-Rad, catalog. no 456–1024) with a Tris/glycine/SDS buffer (Bio-Rad, catalog. no 161–0732). 1/14th the elution volumes were loaded onto the gel for subsequent probing with the anti-CTB antibody for visualization of the molecular weight region encompassing the CTB crosslinked bands, while the remainder of the elutions were loaded for overnight fixation at room temperature in a solution of 5:1:4 ethanol:acetic acid:water. For the immunoblot portion of the gel, the proteins were transferred to a PVDF membrane and probed with a 1:2500 dilution of the anti-CTB antibody (Abcam) in 5% non-fat milk/TBST for 1 hr at room temperature. The blot was then probed with a HRP conjugated secondary antibody (1:5000 dilution) for 1 hr at room temperature, and developed using the SuperSignal West Pico Chemiluminescent Substrate and X-ray film. This blot was then used as an estimate for excision of gel pieces for submission to the UT Southwestern Proteomics Core for analysis.

### In-solution digestion

The membrane fraction of each set of samples (*i.e.* – and + UV) were combined into a 500 µL total volume of "re-suspension buffer" (8 M urea, 50 mM Tris pH 8.0, 0.1% (w/v) RapiGest SF surfactant (Waters, Milford, MA) (catalog no. 186001861), and a protease inhibitor cocktail). Solubilization was initiated by gentle mixing with a pipet tip, followed by incubation on ice for 1 hr; the solubilized

membrane fraction was then transferred to a LoBind microcentrifuge tube (Eppendorf, Newark, NJ) (catalog no. 22431081). The protein content was quantified with a BCA assay kit (Pierce) using BSA as a standard. 600 µg of membrane fraction was diluted to 1.5 µg/µL in re-suspension buffer, and incubated with 25 µL of streptavidin agarose overnight at 4°C with end-over-end rotation. The beads were washed 5 times with 175 µL of re-suspension buffer, 3 times with 2 M urea in DPBS, and 3 times with DPBS. For overnight trypsin digestion, first 25 µL of "digestion buffer"(6 M urea, 100 mM TrisHCl pH 8.0, 2 M thiourea, 50% (v/v) trifluoroethanol, and 0.1% (w/v) RapiGest SF) was added to the beads and incubated at 37°C for 30 min. Next, 2 µL of 200 mM tris (2-carboxyethyl)phosphine (TCEP) (Thermo Scientific, catalog no. PI-20490) was added and incubated at 37°C for 30 min, followed by the addition of 2 µL of 300 mM iodoacetamide (IAA) (Sigma, catalog no. I1149) and incubation at room temperature for 30 min. The reaction was diluted with 450 µL "dilution buffer" (100 mM TrisHCl pH 8.0 and 10 mM CaCl$_2$), and then trypsin (Promega, Madison, WI) (catalog no. V5111) was added to a final concentration of 10 µg/mL and incubated at 37°C overnight (~18 hr) with shaking. Trypsin was then added again to a final concentration of 5 µg/mL and incubated at 37°C for 2 hr with shaking. The RapiGest SF was then cleaved by addition of approximately 0.5% (or 2.5 µL) trifluoroacetic acid (TFA) (Acros Organics, Waltham, MA) (catalog no. AC13972) and incubation at 37°C for 40 min. The samples were centrifuged at 16,200$g$ for 10 min, then the solution transferred to a fresh LoBind microcentrifuge tube. The samples were stored at –20°C and submitted to the UT Southwestern Proteomics Core for analysis.

## Mass spectrometry

Gel pieces were reduced and alkylated with dithiothreitol (DTT) and IAA (Sigma–Aldrich), and digested overnight with trypsin (Promega). Both gel and in-solution samples then underwent solid-phase extraction cleanup with Oasis HLB plates (Waters) and the resulting samples were analyzed by LC/MS/MS, using an Orbitrap Elite (samples from gel) or Q Exactive mass spectrometer (in-solution samples) (Thermo Electron), coupled to identical Ultimate 3000 RSLC-Nano liquid chromatography systems (Dionex). Samples were injected onto a 180 µm i.d., 15 gm long column packed in-house with a reverse-phase material ReproSil-Pur C18-AQ, 3 µm resin (Dr. Maisch GmbH, Ammerbuch-Entringen, Germany), and eluted with a gradient from 1–28% buffer B over 40 min. Buffer A contained 2% (v/v) acetonitrile (ACN) and 0.1% formic acid in water, and buffer B contained 80% (v/v) ACN, 10% (v/v) trifluoroethanol, and 0.08% formic acid in water. The mass spectrometer acquired up to 12 MS/MS spectra (Orbitrap) or 20 MS/MS spectra (Q Exactive) for each full spectrum acquired.

Raw MS data files were converted to a peak list format and analyzed using the central proteomics facilities pipeline (CPFP), version 2.0.3 (*Trudgian and Mirzaei, 2012*). Peptide identification was performed using X!Tandem (*Craig and Beavis, 2004*) and the open MS search algorithm (OMSSA) (*Geer et al., 2004*) against a custom sequence database. CTB (UniProt P01556**)** and mucin sequences (Mucin Database v1.2) (*Hansson, et al., 2015*) were appended to the Human International Protein Index (IPI Human, September 2011) (*Kersey et al., 2004*). Reversed decoy sequences were added for false discovery rate estimation (*Elias and Gygi, 2007*). Fragment and precursor tolerances of 20 ppm and 0.5 Da (Orbitrap) or 0.1 Da (Q Exactive) were specified, and three miscleavages were allowed. Carbamidomethylation of Cys was set as a fixed modification and oxidation of Met was set as a variable modification. Spectral count quantification was performed using the SINQ tool within CPFP (*Trudgian et al., 2011*). Data presented in *Table 1* represent all proteins detected with a spectral count of $\geq$ 3 in the experimental sample (+toxin, +sugar, +UV) and not detected in the control sample (–toxin, +sugar, –UV).

## Immunopurifications

For crosslinking for immunopurification (IP) experiments, 750,000 T84 cells were seeded into 6-cm tissue culture plates in 6 mL of media containing 100 µM Ac$_4$ManNDAz; if required, a glycosylation inhibitor was added at this time. After culturing for 72 hr, the cells were replenished with 1.5 mL of fresh media containing 4 µg/mL CTB (Sigma) (*i.e.*, 6 µg per plate). The toxin was allowed to bind to the cell surface for 45 min at 4°C in the dark. The cells were then either kept at 4°C for an additional 45 min (for the – UV plates) or irradiated on an ice/water bath for 45 min (for + UV plates) at 365 nm. The cells were then washed with DPBS, collected into RIPA lysis buffer with a cell scraper, and

incubated on ice for 30–60 min. The lysate was pelleted at 21,000*g* for 10 min at 4°C to clear the insoluble debris, and the supernatant was retained; protein content was quantified with a BCA assay kit (Pierce) using a BSA standard curve.

For IP with the anti-CEACAM5 and anti-CD44 antibodies, the crosslinked lysates were diluted to 1.5 mg/mL in RIPA buffer, and then 270 µg was added to a 1.5 mL microcentrifuge tube. A 1:50 dilution of normal mouse IgG (EMD Millipore, catalog no. NI03), anti-CEACAM5 antibody, or anti-CD44 antibody was added, and the samples were mixed by end-over-end rotation overnight at 4°C. The lysate/Ab mixture was then added to 10 µL of Protein G sepharose (Sigma, catalog no. P3296) and mixed by end-over-end rotation for 2.5 hr at 4°C. The beads were washed three times with 200 µL RIPA buffer, and eluted with 10 µL 2X SDS loading dye (100 mM TrisHCl pH 6.8, 4% (w/v) SDS, 0.04% (w/v) bromophenol blue, 20% (v/v) glycerol, and 20 mM DTT) for 5 min at 90°C. The supernatant was collected and loaded into a single well of a 7.5% Tris-glycine gel. Two sets of samples were analyzed at a time: the first set to ensure that the IP of the target protein was successful (i.e., probing for CEACAM5 or CD44), and a second set to probe for association of the target protein with CTB. Therefore, after separation and transfer to a PVDF membrane, the blots were probed overnight at 4°C for either CEACAM5 (1:5000 dilution), CD44 (1:2500 dilution), or CTB (1:10,000 dilution) in 5% (w/v) non-fat milk in TBST. The blots were then probed with a HRP conjugated secondary antibody (1:5000 dilution) for 1 hr at room temperature, and developed using the SuperSignal West Pico Chemiluminescent Substrate and X-ray film.

For IP with the anti-CTB antibody, the crosslinked lysates were diluted to 1.5 mg/mL in RIPA buffer, and then 300 µg was added to a 1.5 mL microcentrifuge tube. Then, 0.5 µg of either ant-CTB antibody or normal rabbit IgG (EMD Millipore, catalog no. NI01) was added, and the samples were mixed by end-over-end rotation overnight at 4°C. The lysate/Ab mixture was then added to 10 µL of TrueBlot anti-rabbit Ig IP beads (Rockland Immunochemicals, Limerick, PA) (catalog no. 00-8800-25) and mixed by end-over-end rotation for 2.5 hr at 4°C. The beads were washed four times with 200 µL RIPA buffer, and eluted with 10 µL 2X SDS loading dye for 5 min at 90°C. The supernatant was collected and loaded into a single well of a 6% Tris-glycine gel. After separation and transfer to PVDF membranes, the blots were probed for CTB, CEACAM5, and CD44 as described above.

For co-IP of CTB and CEACAM5 within non-crosslinked lysates, 750,000 T84 cells were seeded in 5 mL of media into 6-cm dishes, and glycosylation inhibitors were added at this time. After culturing for 72 hr, the cells were replenished with 1.5 mL of fresh media containing 4 µg/mL CTB (Sigma) (*i. e.*, 6 µg per plate) and kept at 4°C for 1.25 hr. The cells were then washed with DPBS, collected into RIPA lysis buffer with a cell scraper, and incubated on ice for 30–60 min. The lysate was pelleted at 21,000*g* for 10 min at 4°C to clear the insoluble debris, and the supernatant was retained; protein content was quantified with a BCA assay kit (Pierce) using a BSA standard curve for normalization. For IP with the anti-CTB antibody, the lysates were diluted to 1.4 mg/mL in RIPA buffer, and then 350 µg was added to a 1.5 mL microcentrifuge tube. Then, 0.5 µg of either anti-CTB antibody or normal rabbit IgG was added, and the samples were mixed by end-over-end rotation overnight at 4°C. The lysate/Ab mixture was then added to 10 µL of TrueBlot anti-rabbit Ig IP beads and mixed by end-over-end rotation for 2.5 hr at 4°C. The beads were then washed, eluted, loaded onto a 6% Tris-glycine gel for separation, and probed on a PVDF membrane for CEACAM5 as described previously for the anti-CTB IP of crosslinked samples.

## Measurements of CTB uptake

CTB internalization was measured by the in-cell ELISA, described above, with the following adaptations. The biotin-CTB concentration was 1 µg/ml. During the experiment, control samples (to measure total surface bound biotin-CTB) were washed three times with ice-cold PBS, then kept on ice awaiting analysis. Experimental samples were warmed to 37°C for the indicated times to allow endocytic uptake, then endocytosis was halted by returning cells to ice and washing three times with cold PBS. Non-internalized biotinylated CTB was masked by successive treatment with 50 µg/ml of avidin (Sigma-Aldrich) for 1 hr on ice, followed by three 1-min cold acid washes (0.2 M acetic acid/0.2 M NaCl). Cells were fixed with 4% paraformaldehyde and further permeabilized with 0.1% (v/v) Triton X-100 in PBS. Cells were then incubated at room temperature for 1 hr in streptavidin-HRP conjugate diluted in Q-PBS (PBS supplemented with 0.01% (w/v) saponin, 2% (w/v) BSA, and 0.1% (w/v) lysine,

pH 7.4). Reactive aldehydes and nonspecific binding sites were quenched with Q-PBS. HRP activity was measured as described above.

## Measurement of cAMP

350,000 T84 cells were cultured for 8–10 d in the absence or presence of each glycosylation inhibitor or the corresponding vehicle control, to achieve the following final concentrations: 50 µM NB-DGJ, 1 µg/ml kifunensine, 2 mM benzyl-α-GalNAc, or 100 µM 2F-Fuc. Stock solutions of NB-DGJ and kifunensine were dissolved in water and benzyl-α-GalNAc and 2F-Fuc were dissolved in DMSO. Cells were cultured with inhibitors on 4.67-cm$^2$ Transwell inserts (Costar Laboratories, Cambridge, Mass). Monolayer integrity was evaluated by transepithelial electrical resistance (TEER) measurements, which consistently increased over the 8–10 d culturing period. Typical TEER values for cells treated with the water control, NB-DGJ, and 2F-Fuc were 800–900 $\Omega$ cm$^2$; for cells treated with kifunensine were 700 $\Omega$ cm$^2$; for cells treated with DMSO were 1400 $\Omega$ cm$^2$; for cells treated with benzyl-α-Gal-NAc were 1800 $\Omega$ cm$^2$. Cells were incubated with forskolin (10 µM), VIP (1 µM), or cholera holotoxin (0, 1, or 100 nM) in culture media with 5% $CO_2$ at 37°C for 1 hr, followed 2 washes with cold PBS, then lysed and flash frozen in liquid $N_2$. Accumulated cAMP was measured using Direct Biotrak EIA kit (GE Healthcare), according manufacturer's instructions and normalized by protein content (bicinchoninic acid assay, Pierce BCA protein assay kit, Pierce).

## Analysis of fresh human colon epithelial cells

The study was approved by the Regional Board of Ethics in Medical Research in west Sweden, and all volunteers gave a written informed consent before participation. Five individuals undergoing curative resection of colon tumors at the Sahlgrenska University Hospital were included in the study (3 males and 2 females, aged 67 to 81). Immediately after surgery, a section of the unaffected mucosa was collected from the resection border located at least ten centimeters away from the tumor, and transported in ice-cold PBS before isolation of epithelial cells within less than 2 hr. Tissue specimens were washed with PBS and excess mucus and connective tissue underlying the mucosa was removed. The tissue was cut into 5 mm pieces and epithelial cells released by two cycles of treatment with 1 mM EDTA and 1 mM dithiothreitol (DTT) in HEPES-buffered Hank's balanced salt solution containing 2% of fetal calf serum at 37°C for 15 min with slow stirring (*Lundgren et al., 2005*). Released epithelial cells were collected by filtration, and pooled with epithelial cell fractions from two subsequent EDTA/DTT cycles for use in analysis of CTB binding by flow cytometry. The cells were first stained with Live/Dead aqua (L34957, Invitrogen). After washing, the cells were blocked with various concentrations of AAL-biotin (B-1395, Vector Labs) or kept in buffer prior to addition of antibodies. The antibodies used were anti-CD45 - APC-H7 (clone 2D1, BD), anti-EpCAM - Alexa Flur 488 (clone 9C4, BioLegend, San Diego, CA), and the protein CTB - Alexa Fluor 647 (C34778, Invitrogen). Some samples were stained in the presence of l-fucose (F2252, Sigma-Aldrich), d-galactose (G0750, Sigma-Aldrich), or d-glucose (G8270, Sigma-Aldrich). The staining cocktail was then preincubated for 5 min before addition to the cells. After staining, the cells were washed and analyzed on a flow cytometer (LSR-II, BD) and the results were analyzed using the FlowJo software.

## Acknowledgements

The authors would like to acknowledge the assistance of the UT Southwestern Live Cell Imaging Facility, a Shared Resource of the Harold C Simmons Cancer Center. We thank Lu Zhang, Andres Roig, Jerry Shay, and John Minna (UT Southwestern) for sharing HCEC and HBEC cells, and Babette Weksler (Weill Cornell Medical College), Pierre-Oliver Couraud (INSERM), and Ignacio Romero (The Open University) for sharing hCMEC/D3 cells. We thank Carlos Reis and Sandra Schmid (UT Southwestern) for help with the in-cell ELISA assay. We thank Jan Holmgren (University of Gothenburg), Kim Orth (UT Southwestern), Arun Radhakrishnan (UT Southwestern), and Michael Shiloh (UT Southwestern) for comments on the manuscript.

# Additional information

## Funding

| Funder | Grant reference number | Author |
|---|---|---|
| National Institutes of Health | R01GM090271 | Jennifer J Kohler |
| Welch Foundation | I-1686 | Jennifer J Kohler |
| Cancerfonden | 140393 | Ulf Yrlid<br>Marianne Quiding-Järbrink<br>Bengt Gustavsson |
| Sahlgrenska Universitetssjukhuset | ALFGBG-438071, #165911 | Bengt Gustavsson<br>Ulf Yrlid |
| Danmarks Grundforskningsfond | DNRF107 | Henrik Clausen |
| Hartwell Foundation | Fellowship | Janet E McCombs |
| Cancer Prevention and Research Institute of Texas | RP120613 | Hamid Mirzaei |
| National Institutes of Health | R21AI094427 | Jennifer J Kohler |
| National Institutes of Health | R01MH61345 | Marcel Mettlen |
| National Institutes of Health | F32AI100489 | Amberlyn M Wands |
| National Institutes of Health | T32GM007062 | Andrea C Rodriguez |
| Welch Foundation | I-1850 | Hamid Mirzaei |
| Svenska Forskningsrådet Formas | 68X-22121 | Ulf Yrlid<br>Marianne Quiding-Järbrink |

The funders had no role in study design, data collection and interpretation, or the decision to submit the work for publication.

## Author contributions

AMW, AF, JC, Conception and design, Acquisition of data, Analysis and interpretation of data, Drafting or revising the article; JEMcC, NN, DCT, AL, ST, Acquisition of data, Analysis and interpretation of data, Drafting or revising the article; BD, MM, Acquisition of data, Analysis and interpretation of data; ACR, Analysis and interpretation of data, Contributed unpublished essential data or reagents; MRB, Conception and design, Analysis and interpretation of data, Contributed unpublished essential data or reagents; MQJ, Drafting or revising the article, Contributed unpublished essential data or reagents; BG, Conception and design, Acquisition of data, Contributed unpublished essential data or reagents; CS, HC, Analysis and interpretation of data, Drafting or revising the article, Contributed unpublished essential data or reagents; HM, Conception and design, Analysis and interpretation of data; UY, Conception and design, Acquisition of data, Analysis and interpretation of data, Drafting or revising the article, Contributed unpublished essential data or reagents; JJK, Conception and design, Analysis and interpretation of data, Drafting or revising the article

## Author ORCIDs

Jakob Cervin, http://orcid.org/0000-0002-3840-1008
Jennifer J Kohler, http://orcid.org/0000-0001-5373-3329

## Ethics

Human subjects: This study was performed according to the Declaration of Helsinki and approved by the Regional Board of Ethics in Medical Research in West Sweden, approval no 249-15. Patients received oral and written information about the study by the study nurse the day before surgery, and if they agreed to participate, they signed a consent form stating permission to use the tissue and publish the results in a way that did not reveal the identity of the donor.

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
