## [Decision Letter]

Thank you for submitting your work entitled "Fucosylation and protein glycosylation create functional receptors for cholera toxin" for peer review at *eLife*. Your submission has been favorably evaluated by Randy Schekman (Senior Editor) and three reviewers, one of whom is a member of our Board of Reviewing Editors.

The reviewers have discussed the reviews with one another and the Reviewing Editor has drafted this decision to help you prepare a revised submission.

In this report, the authors challenge the widely accepted view that ganglioside GM1 is the glycan ligand of cholera toxin (CT), enabling CT to enter and intoxicate intestinal cells. GM1 is indeed the highest affinity glycan ligand of CT, and is known to support CTA entry into host cells in model systems. But the apparent lack of this ligand on gut epithelial cells has raised the possibility that other weaker avidity ligands mediate CT entry and intoxication. Here, using a method involving metabolic labeling of cells to contain UV reactive sialic acids that crosslink glycan ligands to CT, the authors present compelling evidence that glycoproteins can bind CTB on the surface of the intestinal epithelial cell lines and mediate toxin uptake by endocytosis. There is, however, incomplete evidence that the glycoprotein receptors can mediate intoxication. Clear evidence is presented for fucosylated ligands to mediate CTB binding, and depending on the intestinal cell line O-linked glycans (T84) or both O-linked and N-linked glycans (Colo205) can carry ligands for CT.

The manuscript from Wands et al. investigates binding of cholera toxin to a number of different cell lines, some of which do not express detectable amounts of GM1, the most widely recognized receptor for cholera toxin. The authors cite previous studies indicating that GM1 may not be the sole cell surface receptor for cholera toxin. However, these alternative glycoprotein receptors have not been characterized, nor is it known whether N-linked or O-linked oligosaccharides are the ligands that bind the toxin. The authors use a diverse set of experimental tools (photoactivatable sugars, glycosylation inhibitors, lectins and monosaccharides, etc.) to document binding of cholera toxin to N-linked and O-linked glycans on cell surface glycoproteins, apparently via terminal fucose residues. The effort to extend this conclusion to normal gut epithelia yielded results that were less conclusive. While fucosylated protein-linked oligosaccharides may contribute to cholera toxin binding, they do not appear to be the only class of receptor. Overall, this manuscript makes a significant contribution towards characterizing glycoprotein receptors for cholera toxin on human cells.

Essential revisions:

1) The text does not mention extensive glycan array data obtained by the Consortium for Functional Glycomics by several investigators, including detailed studies titrating CT on the array from nanogram to 100 microgram (The data can be accessed at http://www.functionalglycomics.org/glycomics). This data shows that fucosylated ligands are lower avidity ligands of CT. This glycan array data has not been published, so it does not detract from this manuscript's evidence that terminal fucose residues contribute to cholera toxin binding. The glycan array data has been presented at numerous public meetings as an example of the power of glycan arrays to detect lower avidity ligands. It would be good to cite the glycan array data in some way since it supports the use of fucosyltransferase inhibitors.

2) The use of Ac_4_ManNDAz that introduces a photoactivatable group into cell surface sialic acids implies that the glycan ligands also contain sialic acid, otherwise, how do you account for crosslinking? But the authors conclude that sialic acids are not involved? The inability of lectins to block CT binding is a negative experiment, and is weak evidence at best that sialic acids are not involved. Have the authors tried the sialyltransferase inhibitor 3F-NeuAc? It is likely that the functional ligand of CT contains both sialic acid and fucose?

3) The text often includes statements that overstate the results. For example the text states that fucose is critical for cholera toxin binding, in reference to an experiment where a fucose inhibitor causes a 2-fold reduction in binding activity (Figure 5). Given that binding of UEA-1 is entirely blocked, the authors need to consider the possibility that fucose-deficient oligosaccharides may also be recognized by CT. In this regard, the authors should test whether galactose has inhibitor activity (Figure 6). Also, GM1 does not contain fucose.

4) The authors need to provide additional information about the internalization experiments shown in Figure 7 and Figure 7—figure supplement 1. The confusing issue here is the effect of inhibitors on total cell-surface binding of CT (Figure 7—figure supplement 1). A previous figure (Figure 5) showed that 2F-Fuc causes a ∼2-fold reduction in CT binding to T84 cells and Colo205 cells. Benzyl-α-GalNAc treatment also caused a ∼2-fold reduction in CT binding (Figure 3). Figure 7—figure supplement 1, shows that these compounds cause only a mild reduction in cell surface binding, but instead reduce internalization. Were the earlier binding assays (Figure 3 and Figure 5) done under conditions that allowed internalization? If so, this distinction needs to be clearly stated in the manuscript. Does the experiment in Figure 7 only monitor internalization of CT molecules that are not bound to fucosylated proteins (+2F-Fuc)?

5) The evidence against the presence of GM1 in intestinal cell membranes is weak. This is an important point for the paper's central thesis as it underlies the interpretation of the functional studies for the glycoprotein receptors for CTB. Several studies show GM1 in intestinal cells (Critchley et al. 1981, J Biol Chem 256: 8724-8731 as an example), so the lack of detection in this paper is unexpected. The result might be explained by the low level of GM1 in intestinal cells coupled with the relatively insensitive approach used in this paper to detect the gangliosides (resorcinol stain). The authors should re-examine these studies by making use of "far-western" toxin blots of TLC plates as described (Magnani et al. 1980, Anal Biochem 109:399-402 and Critchley et al. 1981, J Biol Chem 256: 8724-8731 as examples).

6) Are cell surface glycoproteins functional receptors for CT? This would be a most important discovery with major implications to the fields of toxin and cell and membrane biology. It would imply that a cell surface membrane protein can traffic retrograde through the endosome/Golgi network all the way to the ER – a very unusual event (I cannot think of an example). Almost all toxins and viruses that use the ER to enter host cells bind glycosphingolipids. Ricin, an "apparent possible" exception, binds highly promiscuously to membrane components containing galactose – so its trafficking receptor is not well defined and may well be the glycosphingolipids.

7) The evidence that glycoproteins are acting as functional trafficking receptors leading to toxicity for cholera toxin is also weak. It is based entirely on use of the chemical inhibitors of glycan synthesis which undoubtedly have off-target effects. And the experiment that tests the idea lacks controls for off-target inhibition of adenylyl cyclase or its coupling to Gs (a frequent event with chemical inhibitors). These studies need to be reproduced using the Gs agonist vasointestinal peptide (VIP) and forskolin to measure maximal adenylyl cyclase activity under each condition – and compared directly with the toxin-induced signals (or calculated as a ratio).

It is also necessary to explain the result in Figure 7 that NB-DGJ inhibits CT toxicity in intestinal cells – what is explanation for this if the cells lack GM1 gangliosides?

8) The intoxication experiments included several puzzling results. Why does kifunensine treatment enhance internalization, but reduce intoxication? Why does NB-DGJ treatment reduce intoxication if the T84 cells do not synthesize detectable amounts of gangliosides including GM1? The possibility that multiple steps in intoxication are sensitive to perturbation of oligosaccharide biosynthesis weakens the conclusion that fucosylated oligosaccharides are the most critical receptor.

9) Examine the intestinal cell lines for GM1 by ligand blot of TLC plates using radiolabeled or HRP-labeled CT beta-subunit or other mass spectrometry methods after treatment with ganglioside inhibitors(s) (NB-DGJ) to show complete loss of GM1. Test for toxicity (or retrograde trafficking into the ER) after showing complete loss of GM1 expression by treatment with NB-DGJ (or other inhibitor of ganglioside sphingolipid synthesis). Toxicity in those intestinal cells lacking GM1 should also be shown to be sensitive to brefeldin A (will block retrograde transport to ER and provide additional control unrelated to GM1).

It is also necessary to explain the result in Figure 7 that NB-DGJ inhibits CT toxicity in intestinal cells – what is explanation for this if the cells lack GM1 gangliosides?

---

## [Author Response]

In this report, the authors challenge the widely accepted view that ganglioside GM1 is the glycan ligand of cholera toxin (CT), enabling CT to enter and intoxicate intestinal cells. GM1 is indeed the highest affinity glycan ligand of CT, and is known to support CTA entry into host cells in model systems. But the apparent lack of this ligand on gut epithelial cells has raised the possibility that other weaker avidity ligands mediate CT entry and intoxication. Here, using a method involving metabolic labeling of cells to contain UV reactive sialic acids that crosslink glycan ligands to CT, the authors present compelling evidence that glycoproteins can bind CTB on the surface of the intestinal epithelial cell lines and mediate toxin uptake by endocytosis. There is, however, incomplete evidence that the glycoprotein receptors can mediate intoxication. Clear evidence is presented for fucosylated ligands to mediate CTB binding, and depending on the intestinal cell line O-linked glycans (T84) or both O-linked and N-linked glycans (Colo205) can carry ligands for CT.

We appreciate that the reviewers found compelling the evidence that CTB binds to glycoproteins, and these binding events can mediate toxin internalization. To solidify the evidence that fucosylated receptors play a role in host cell intoxication, we performed additional control experiments to assess off-target effects of inhibitors on adenylate cyclase activity and the effects of brefeldin A on host cell intoxication. See responses 7, 8, and 9 below for more details.

The manuscript from Wands et al. investigates binding of cholera toxin to a number of different cell lines, some of which do not express detectable amounts of GM1, the most widely recognized receptor for cholera toxin. The authors cite previous studies indicating that GM1 may not be the sole cell surface receptor for cholera toxin. However, these alternative glycoprotein receptors have not been characterized, nor is it known whether N-linked or O-linked oligosaccharides are the ligands that bind the toxin. The authors use a diverse set of experimental tools (photoactivatable sugars, glycosylation inhibitors, lectins and monosaccharides, etc.) to document binding of cholera toxin to N-linked and O-linked glycans on cell surface glycoproteins, apparently via terminal fucose residues. The effort to extend this conclusion to normal gut epithelia yielded results that were less conclusive. While fucosylated protein-linked oligosaccharides may contribute to cholera toxin binding, they do not appear to be the only class of receptor. Overall, this manuscript makes a significant contribution towards characterizing glycoprotein receptors for cholera toxin on human cells.

We appreciate that the reviewers found our contribution to characterizing glycoprotein receptors for cholera toxin to be significant. Based on the reviewers’ comments, we focused our revision on the following points: 1) using more sensitive detection methods to confirm that GM1 levels in the intestinal epithelial cell lines were indeed low and to validate that the NB-DGJ concentration we chose to use for our assays was effective, 2) assessing contributions that sugars other than L-fucose play in glycoprotein recognition by CTB, and 3) evaluating off-target effects that glycosylation inhibitors exert on cAMP accumulation.

1) The text does not mention extensive glycan array data obtained by the Consortium for Functional Glycomics by several investigators, including detailed studies titrating CT on the array from nanogram to 100 microgram (The data can be accessed at http://www.functionalglycomics.org/glycomics). This data shows that fucosylated ligands are lower avidity ligands of CT. This glycan array data has not been published, so it does not detract from this manuscript's evidence that terminal fucose residues contribute to cholera toxin binding. The glycan array data has been presented at numerous public meetings as an example of the power of glycan arrays to detect lower avidity ligands. It would be good to cite the glycan array data in some way since it supports the use of fucosyltransferase inhibitors.

We thank the reviewers for this comment encouraging us to acknowledge the CFG microarray data. Indeed, examination of the CFG microarray data played an essential role in our decision to test whether fucose contributes to cholera toxin binding. We omitted this fact in the original submission because the CFG data contains some conflicting results for different trials of CTB binding. Nonetheless, we now acknowledge the essential role of the CFG data by citing the CFG website in the revised submission.*2) The use of Ac_4_ManNDAz that introduces a photoactivatable group into cell surface sialic acids implies that the glycan ligands also contain sialic acid, otherwise, how do you account for crosslinking? But the authors conclude that sialic acids are not involved? The inability of lectins to block CT binding is a negative experiment, and is weak evidence at best that sialic acids are not involved. Have the authors tried the sialyltransferase inhibitor 3F-NeuAc? It is likely that the functional ligand of CT contains both sialic acid and fucose?*

The reviewers point out that the SiaDAz crosslinking data imply that CTB is capable of recognizing sialylated glycoproteins, raising the question of whether sialic acid is required for CTB recognition of glycoproteins. With respect to the crosslinking data, one possible interpretation is that sialic acid is not required, but can fortuitously capture some interactions through proximity to the preferred fucosylated ligand. Thus, additional experiments are required to assess the role of sialylation. As suggested, we used the sialyltransferase inhibitor 3F-NeuAc to assess whether sialylation is required for CTB binding to Colo205 cells. (Note that we cannot use 3F-NeuAc in crosslinking experiments because it would be expected to block SiaDAz incorporation.) We observed that inhibition of sialylation resulted in slightly *increased* binding to CTB, consistent with a slight increase in binding of a fucose-recognizing lectin that also occurs with inhibition of sialylation (new data shown in Figure 8 and Figure 8—figure supplement 2). We also attempted to use 3F-NeuAc to assess the role of sialylation in CTB binding to T84 cells, but did not observe inhibition of sialylation at the 3F-NeuAc concentrations tested. Finally, we tested whether free Neu5Ac could inhibit binding to colonic epithelial cell lines. We found that Neu5Ac strongly inhibited CTB binding to T84 cells (new data in Figure 6), but only slightly inhibited CTB binding to Colo205 cells (new data in Figure 6). Based on these data and the lectin data shown in the previous submission (now shown in Figure 7), we cannot make a definitive conclusion regarding a role for sialic acid in CTB binding to colonic epithelial cell lines. Since the role of sialylation is not central to the main conclusions of the current manuscript, we chose to curtail these experiments and moderated our language in the manuscript regarding the role of sialylation in binding of CTB to glycoproteins in these cell lines. We will continue to pursue understanding the role of sialylation in future work.

As a side note, the fact that inhibition of sialylation results in increased binding of CTB to Colo205 cells is an additional piece of evidence suggesting that GM1 is not the dominant CTB ligand on Colo205 cells.

3) The text often includes statements that overstate the results. For example the text states that fucose is critical for cholera toxin binding, in reference to an experiment where a fucose inhibitor causes a 2-fold reduction in binding activity (Figure 5). Given that binding of UEA-1 is entirely blocked, the authors need to consider the possibility that fucose-deficient oligosaccharides may also be recognized by CT. In this regard, the authors should test whether galactose has inhibitor activity (Figure 6). Also, GM1 does not contain fucose.

We edited the manuscript to modulate statements about the role of fucose in CTB binding. We wished to emphasize the fact the 2F-Fuc inhibitor caused a substantial decrease in CTB binding, but did not intend to imply that every CTB ligand contains fucose. Indeed, we agree with the reviewers’ proposal that multiple ligands, including fucose-deficient glycans, could contribute to CTB binding. In line with the reviewers’ suggestion, we tested additional monosaccharides to attempt to identify other structures recognized by CTB. For Colo205 cells, fucose was the most potent inhibitor. Galactose and Neu5Ac inhibited moderately, while glucose, GlcNAc, and GalNAc did not inhibit CTB binding (new data in Figure 6). For T84 cells, both fucose and Neu5Ac were strong inhibitors, galactose inhibited slightly, and glucose, GlcNAc, and GalNAc did not inhibit (new data in Figure 6). Finally, for primary intestinal epithelial cells, fucose inhibits CTB binding, but glucose and galactose do not inhibit (new data in Figure 10). Thus, fucose was the only monosaccharide that potently inhibited CTB binding to both colonic epithelial cell lines and to primary cells, distinguishing this cell type from the Jurkat cell line where fucose does not inhibit (Figure 6). Other monosaccharides, namely Neu5Ac and galactose, showed variable and/or weaker inhibition on colonic epithelial cells, a fact that we now point out in the text.

4) The authors need to provide additional information about the internalization experiments shown in Figure 7 and Figure 7—figure supplement 1. The confusing issue here is the effect of inhibitors on total cell-surface binding of CT (Figure 7—figure supplement 1). A previous figure (Figure 5) showed that 2F-Fuc causes a ∼2-fold reduction in CT binding to T84 cells and Colo205 cells. Benzyl-α-GalNAc treatment also caused a ∼2-fold reduction in CT binding (Figure 3). Figure 7—figure supplement 1, shows that these compounds cause only a mild reduction in cell surface binding, but instead reduce internalization. Were the earlier binding assays (Figure 3 and Figure 5) done under conditions that allowed internalization? If so, this distinction needs to be clearly stated in the manuscript. Does the experiment in Figure 7 only monitor internalization of CT molecules that are not bound to fucosylated proteins (+2F-Fuc)?

We thank the reviewers for calling our attention to Figure 7, which allowed us to identify a problem in the data reported in the initial submission, as outlined below. First, to be clear, the binding assays reported in the earlier figures (formerly Figure 3 and Figure 5; now Figure 4 and Figure 8 in the revised manuscript) were performed at 4°C, conditions that did not allow CTB internalization. The problem we identified was in the data initially reported as Figure 7—figure supplement 1, showing the total binding of CTB to T84 cells, as measured by ELISA. The incubation times used for this experiment resulted in strong OPD signals that fell outside the linear range of detection. These skewed values diminished the differences in CTB binding for the different glycosylation inhibitors. Further, these values for total CTB binding were also used, together with measurements of internalized CTB, to calculate internalization efficiencies, shown in the previous version of Figure 7. Thus, the internalization efficiency plots were also inaccurate. In the revision, we have modified the experimental protocol to avoid saturating OPD signals, so that we accurately measure total CTB binding in this experiment. These results are shown in Figure 9—figure supplement 1 and are now consistent with the results shown in Figure 4 and Figure 8. We now report the amount of internalized CTB in the main Figure 9. Notably, little to no CTB was internalized when either fucosylation or O-linked glycosylation was inhibited. In answer to the reviewers’ question, any CTB that is internalized when cells are cultured with 2F-Fuc could reflect CTB molecules that are not bound to fucosylated proteins, or possibly CTB bound to a small amount of fucosylated proteins that might remain present even with 2F-Fuc inhibition. Finally, in the revised manuscript, we chose not to show calculated internalization efficiencies (internalized CTB divided by total CTB bound), out of concern that the low amount of total CTB bound under some conditions could comprise the accuracy of these calculations. Based on these results, the main conclusions for internalization experiments are that inhibition of fucosylation and of O-linked glycosylation each result in a dramatic reduction in the amount of CTB that enters cells, and that NB-DGJ treatment does not substantially affect CTB internalization. The text has been modified to more clearly state these conclusions.*5) The evidence against the presence of GM1 in intestinal cell membranes is weak. This is an important point for the paper's central thesis as it underlies the interpretation of the functional studies for the glycoprotein receptors for CTB. Several studies show GM1 in intestinal cells (Critchley et al. 1981, J Biol Chem, 256: 8724-8731 as example), so the lack of detection in this paper is unexpected. The result might be explained by the low level of GM1 in intestinal cells coupled with the relatively insensitive approach used in this paper to detect the gangliosides (resorcinol stain). The authors should re-examine these studies by making use of "far-western" toxin blots of TLC plates as described (Magnani et al. 1980, Anal Biochem 109:399-402 and Critchley et al. 1981, J Biol Chem 256: 8724-8731 as examples).*

As suggested by the reviewers, we pursued the more sensitive technique of using ^125^I-labeled CTB to probe TLC plates in an attempt to detect GM1 in Colo205 and T84 cells. To perform these experiments, we enlisted the help of Susann Teneberg (University of Gothenburg), an expert in the practice of this technique specifically and in glycolipid analysis in general. Even with the use of this sensitive detection method, GM1 was not detectable in either Colo205 or T84 extracts (new data in Figure 3). These experiments do not formally exclude the possibility that GM1 may be present in Colo205 or T84 cells, but they greatly reduce the upper limit for the amount of GM1 in the cell lines.

TLCs of the extracts were also probed with ^125^I-labeled *E. cristagalli* lectin, revealing a single reactive species in T84 cells, likely corresponding to neolactotetraosylceramide (Galβ4GlcNAcβ3Galβ4Glcβ1Cer; new data in Figure 3). This band was absent from T84 cells cultured with NB-DGJ, confirming the effectiveness of the NB-DGJ inhibitor at blocking production of glucosylceramide glycolipids. Thus, even if some GM1 is present in T84 cells, it is reasonable to assume that this level is further reduced by culturing the cells with NB-DGJ.

6) Are cell surface glycoproteins functional receptors for CT? This would be a most important discovery with major implications to the fields of toxin and cell and membrane biology. It would imply that a cell surface membrane protein can traffic retrograde through the endosome/Golgi network all the way to the ER – a very unusual event (I cannot think of an example). Almost all toxins and viruses that use the ER to enter host cells bind glycosphingolipids. Ricin, an "apparent possible" exception, binds highly promiscuously to membrane components containing galactose – so its trafficking receptor is not well defined and may well be the glycosphingolipids.

We propose that cell surface glycoproteins are functional CT receptors in the sense that they enable CT to enter the cell via a pathway that leads to host cell intoxication. However, we recognize that the mechanism of host cell intoxication by CT includes multiple steps. The data we report in the manuscript demonstrate (1) that fucosylated glycoproteins mediate a large fraction of CTB binding to T84 and Colo205 cells, (2) that fucosylated glycoproteins play an important role in the mechanism by which CTB enters T84 cells, and (3) that entry of CT into T84 cells via a fucose-dependent mechanism is on-pathway to host cell intoxication. However, these data do not exclude the possibility that CT might also enter cells via additional mechanisms, a point that we attempted to make more clearly in the revised manuscript. Further, once CT enters cells, we do not know whether it remains bound to fucosylated glycoproteins. CT might be handed off to other molecules for further trafficking steps. Indeed, GM1 could play role in these later trafficking steps, consistent with our observation that NB-DGJ treatment does not affect CTB entry, but does reduce host cell intoxication. In addition, others have reported that the C-terminal KDEL on CTA can enhance the efficiency of host cell intoxication through interactions with the KDEL receptor that mediate retrograde trafficking (for example, see J. Cell Biol. 131: 951-962), providing a potential mechanism for CT to traffic to the ER. We revised the manuscript to make clear that the data reported in this manuscript do not report directly on the trafficking steps that occur after CT enters the host cell, and to point out that CT might exploit multiple parallel pathways for host cell intoxication. We plan to explore whether fucosylated glycoproteins remain associated with CT throughout later trafficking steps, but feel that these studies are beyond the scope of the current manuscript.*7) The evidence that glycoproteins are acting as functional trafficking receptors leading to toxicity for cholera toxin is also weak. It is based entirely on use of the chemical inhibitors of glycan synthesis which undoubtedly have off-target effects. And the experiment that tests the idea lacks controls for off-target inhibition of adenylyl cyclase or its coupling to Gs (a frequent event with chemical inhibitors). These studies need to be reproduced using the Gs agonist vasointestinal peptide (VIP) and forskolin to measure maximal adenylyl cyclase activity under each condition – and compared directly with the toxin-induced signals (or calculated as a ratio).*

It is also necessary to explain the result in Figure 7 that NB-DGJ inhibits CT toxicity in intestinal cells – what is explanation for this if the cells lack GM1 gangliosides?

We thank the reviewers for bringing up this point, as the suggested experiments helped to clarify interpretation of our data. As proposed, we tested whether the inhibitors of glycosylation affected activation of adenylyl cyclase activity by VIP and forskolin. Both kifunensine and benzyl-α-GalNAc caused reduced activation of adenylyl cyclase activity by both VIP and forskolin (new data in Figure 9—figure supplement 2). Thus, we feel that we cannot interpret the effects of kifunensine and benzyl-α-GalNAc on cholera toxin-induced cAMP production and we have moved these results to Figure 9—figure supplement 2. In contrast, neither NB-DGJ nor 2F-Fuc affected activation of adenylyl cyclase activity by either VIP or forskolin, but both of these inhibitors do cause reduced CT-induced cAMP production, with 2F-Fuc causing a more dramatic effect (Figure 9). Since 2F-Fuc treatment also causes reduced CTB binding and cell entry, the most likely explanation for this result is that reduced fucosylation leads to less CT entering cells, and thereby reduces host cell intoxication. Since NB-DGJ treatment does not grossly affect CTB binding or internalization, we think it is reasonable to consider three possible explanations for the effect that NB-DGJ exerts on CT-induced cAMP production. One possibility is that the cells contain a level of GM1 that is not detectable by the sensitive methods employed in Figure 3, but which is nonetheless capable of mediating host cell intoxication via a pathway that operates either in parallel or in concert with that mediated by fucosylated glycoproteins. A second possibility is that GM1 is not required for the initial steps of CT binding and internalization, but is required for later steps in intoxication (i.e. the potential hand-off mechanism discussed in our response to item #6 above). A final possibility is that GM1 is not essential, but other glucosylceramide glycolipids, which are also reduced due to NB-DGJ treatment, play roles in the trafficking and intoxication process. We now enumerate these possibilities explicitly in the manuscript. In summary, with the addition of these control experiments, the data now clearly demonstrate a role for fucosylation in the mechanism of host cell intoxication by CT.

8) The intoxication experiments included several puzzling results. Why does kifunensine treatment enhance internalization, but reduce intoxication? Why does NB-DGJ treatment reduce intoxication if the T84 cells do not synthesize detectable amounts of gangliosides including GM1? The possibility that multiple steps in intoxication are sensitive to perturbation of oligosaccharide biosynthesis weakens the conclusion that fucosylated oligosaccharides are the most critical receptor.

As discussed above in response to item #7, we now show that kifunensine exerts a general effect that reduces adenylyl cyclase activation, which is not specific to host cell intoxication by CT and that there are several potential explanations for the result that NB-DGJ reduces the severity of host cell intoxication. To be clear, we have not excluded the possibility that GM1 can function as a receptor. Rather, we provide positive evidence that fucosylated glycoproteins mediate CTB binding and internalization and, further, that fucose-dependent internalization of CT is on-pathway to host cell intoxication.

9) Examine the intestinal cell lines for GM1 by ligand blot of TLC plates using radiolabeled or HRP-labeled CT beta-subunit or other mass spectrometry methods after treatment with ganglioside inhibitors(s) (NB-DGJ) to show complete loss of GM1. Test for toxicity (or retrograde trafficking into the ER) after showing complete loss of GM1 expression by treatment with NB-DGJ (or other inhibitor of ganglioside sphingolipid synthesis). Toxicity in those intestinal cells lacking GM1 should also be shown to be sensitive to brefeldin A (will block retrograde transport to ER and provide additional control unrelated to GM1).

As suggested, we used ^125^I-labeled CTB to probe Colo205 and T84 extracts separated by TLC, but were unable to detect GM1 in either cell line (new data in Figure 3). We did demonstrate the effectiveness of the NB-DGJ inhibitor by showing that it reduced that amount of another glucosylceramide glycolipid recognized by *E. cristagalli* lectin (new data in Figure 3). Thus, if there is a small amount of GM1 in T84 cells, we expect that the level will be reduced by NB-DGJ treatment. In new data presented in Figure 9—figure supplement 4, we show that CT toxicity in control cells is sensitive to brefeldin A. Similarly, in cells that were cultured with NB-DGJ and treated with brefeldin A, the amount of CT-induced cAMP production is equivalent to the background level observed with no CT treatment. Since the magnitude of CT-induced cAMP production in the NB-DGJ-treated cells is small, the difference attributable to brefeldin A has a p value of 0.06. Despite the borderline significance of the p value, the data are most consistent with the residual CT intoxication in NB-DGJ-treated cells occurring through mechanism that depends on the secretory pathway.

As discussed in the response to item #7, there are several possible explanations for the fact that NB-DGJ inhibits CT toxicity, including the possibility that there is GM1 in these cells that cannot be detected by the sensitive methods employed in Figure 3, but that functions as a receptor for some fraction of the CT. We now discuss these possibilities explicitly in the manuscript. We emphasize that we have not excluded a role for GM1 in host cell intoxication by CT, rather our data provide positive evidence that fucosylation and glycoproteins play significant roles.